# Learning Controllable Adaptive Simulation for Multi-resolution Physics

**Tailin Wu**[1*] **Takashi Maruyama**[1,2*]**, Qingqing Zhao**[1*]**, Gordon Wetzstein**[1]**, Jure Leskovec**[1]
[1]Stanford University,  [2]NEC Corporation
`tailin@cs.stanford.edu,49takashi@nec.com,cyanzhao@stanford.edu,`
`gordonwz@stanford.edu,jure@cs.stanford.edu`

## Abstract

Simulating the time evolution of physical systems is pivotal in many scientific and engineering problems. An open challenge in simulating such systems is their multi-resolution dynamics: a small fraction of the system is extremely dynamic, and requires very fine-grained resolution, while a majority of the system is changing slowly and can be modeled by coarser spatial scales. Typical learning-based surrogate models use a uniform spatial scale, which needs to resolve to the finest required scale and can waste a huge compute to achieve required accuracy. In this work, we introduce Learning controllable Adaptive simulation for Multi-resolution Physics (LAMP) as the first full deep learning-based surrogate model that jointly learns the evolution model and optimizes appropriate spatial resolutions that devote more compute to the highly dynamic regions. LAMP consists of a Graph Neural Network (GNN) for learning the forward evolution, and a GNN-based actor-critic for learning the policy of spatial refinement and coarsening. We introduce learning techniques that optimizes LAMP with weighted sum of error and computational cost as objective, allowing LAMP to adapt to varying relative importance of error vs. computation tradeoff at inference time. We evaluate our method in a 1D benchmark of nonlinear PDEs and a challenging 2D mesh-based simulation. We demonstrate that our LAMP outperforms state-of-the-art deep learning surrogate models, and can adaptively trade-off computation to improve long-term prediction error: it achieves an average of 33.7% error reduction for 1D nonlinear PDEs, and outperforms MeshGraphNets + classical Adaptive Mesh Refinement (AMR) in 2D mesh-based simulations. Project website with data and code can be found at: `http://snap.stanford.edu/lamp`.

## 1 Introduction

Simulating the time evolution of a physical system is of vital importance in science and engineering (Lynch, 2008; Carpanese, 2021; Sircombe et al., 2006; Courant et al., 1967; Lelievre & Stoltz, 2016). Usually, the physical system has a multi-resolution nature: a small fraction of the system is highly dynamic, and requires very fine-grained resolution to simulate accurately, while a majority of the system is changing slowly. Examples include hazard prediction in weather forecasting (Majumdar et al., 2021), disruptive instabilities in the plasma fluid in nuclear fusion (Kates-Harbeck et al., 2019), air dynamics near the boundary for jet engine design (Athanasopoulos et al., 2009), and more familiar examples such as wrinkles in a cloth (Pfaff et al., 2021) and fluid near the boundary for flow through the cylinder (Vlachas et al., 2022). Due to the typical huge size of such systems, it is pivotal that those systems are simulated not only *accurately*, but also with as small of a *computational cost* as possible. A uniform spatial resolution that pays similar attention to regions with vastly different dynamics, will waste significant compute on slow-changing regions while may be insufficient for highly dynamic regions.

To accelerate physical simulations, deep learning (DL)-based surrogate models have recently emerged as a promising alternative to complement (Um et al., 2020) or replace (Li et al., 2021) classical solvers. They reduce computation and accelerate the simulation with larger spatial (Um

---

[*]Equal contribution.

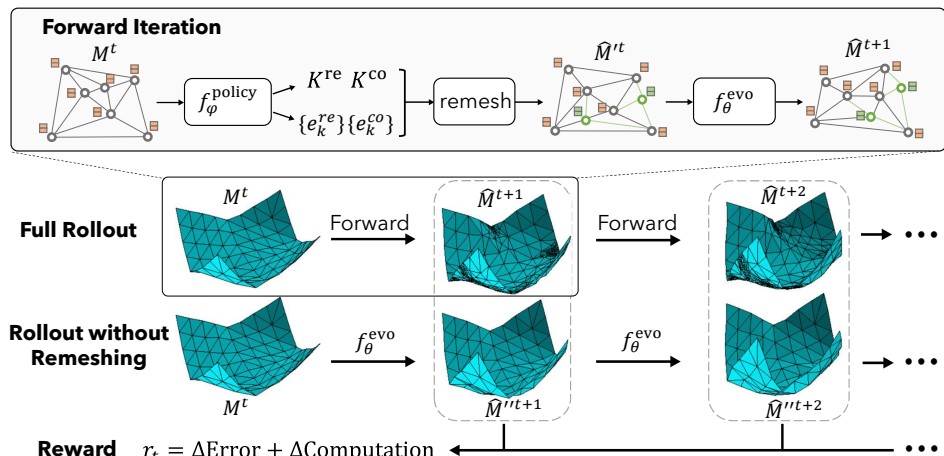

Figure 1: LAMP schematic. The forward iteration (upper box) first uses the policy $f_\varphi^{\text{policy}}$ to decide the number $K^{\text{re}}$ and $K^{\text{co}}$ of edges as well as which edges among the full mesh to be refined or coarsened, and then executes remeshing and interpolation. The evolution model $f_\theta^{\text{evo}}$ is applied to the updated mesh $\hat{M}'^t$ to predict the state $\hat{M}^{t+1}$ at the next time step. We use the *reduction* of both *Error* and *Computation* (mesh size), compared to a multi-step rollout without remeshing, as reward to learn the policy. For more details, see Section 3.2.

et al., 2020; Kochkov et al., 2021) or temporal resolution (Li et al., 2021), or via latent representations (Sanchez-Gonzalez et al., 2020; Wu et al., 2022). However, current deep learning-based surrogate models typically assume a uniform or fixed spatial resolution, without *learning* how to best assign computation to the most needed spatial region. Thus, they may be insufficient to address the aforementioned multi-resolution challenge. Although adaptive methods, such as Adaptive Mesh Refinement (AMR) (Soner et al., 2003; Cerveny et al., 2019) exist for classical solvers, they share similar challenge (*e.g.,* slow) as classical solvers. A deep learning-based surrogate models, that is able to learn both the evolution and learn to assign computation to the needed region, is needed.

In this work, we introduce Learning controllable Adaptive simulation for Multi-resolution Physics (LAMP) as the first fully DL-based surrogate model that jointly learns the evolution model and optimizes appropriate spatial resolutions that devote more compute to the highly dynamic regions. Our key insight is that by explicitly setting the error and computation as the combined objective to optimize, the model can learn to adaptively decide the best local spatial resolution to evolve the system. To achieve this goal, LAMP consists of a Graph Neural Network (GNN)-based evolution model for learning the forward evolution, and a GNN-based actor-critic for learning the policy of discrete actions of local refinement and coarsening of the spatial mesh, conditioned on the local state and a coefficient $\beta$ that weights the relative importance of error vs. computation. The policy (actor) outputs both the *number* of refinement and coarsening actions, and *which* edges to refine or coarsen, while the critic evaluates the expected reward of the current policy. The full system is trained with an alternating fashion, iterating between training the evolution model with supervised loss, and training the actor-critic via reinforcement learning (RL). Taken together, a single instance of evolution model and actor-critic jointly optimizes reduction of error and computation for the physical simulation, and can operate across the relative importance of the two metrics at inference time.

We evaluate our model on a 1D benchmark of nonlinear PDEs (which tests generalization across PDEs of the same family), and a challenging 2D mesh-based simulation of paper folding. In 1D, we show that our model outperforms state-of-the-art deep learning-based surrogate models in terms of long-term evolution error by 33.7%, and can adaptively tradeoff computation to improve long-term prediction error. On a 2D mesh-based simulation, our model outperforms state-of-the-art Mesh-GraphNets + classical Adaptive Mesh Refinement (AMR) in 2D mesh-based simulations.

## 2  PROBLEM SETTING AND RELATED WORK

We consider the numerical simulation of a physical system, following the notation introduced in (Pfaff et al., 2021). The system's state at time $t$ is discretized into the mesh-based state $M^t =$

$(V^t, E^t), t = 0, 1, 2, ...$, where $E^t$ is the mesh edges and $V^t$ is the states at the nodes at time $t$. Each node $i \in V$ contains the mesh-space coordinate $u_i$ and dynamic features $q_i$. Note that this representation of the physical system is very general. It includes Eulerian systems (Wu et al., 2022) where the mesh is fixed and the field $q_i$ on the nodes are changing, and Lagrangian systems (Sanchez-Gonzalez et al., 2020; Pfaff et al., 2021) where the mesh coordinate in physical space is also dynamically moving (for this case, an additional world coordinate $x_i$ is accompanying the mesh coordinate $u_i$). During prediction, a simulator $f$ (classical or learned) autoregressively predicts system's state $\hat{M}^{t+1}$ at the next time step based on its previous prediction $\hat{M}^t$ at previous time step:

$$\hat{M}^{t+1} = f(\hat{M}^t), t = 0, 1, 2, ... \tag{1}$$

where $\hat{M}^0 = M^0$ is the initial state. During the prediction, both the dynamic features $V^t$ at the mesh nodes, and the mesh topology $E^t$ can be changing. The error is typically computed by comparing the prediction and ground-truth after long-term prediction: error $:= \ell(\hat{M}^t, M^t)$ for a metric $\ell$ (*e.g.,* MSE, RMSE), and the computation cost (in terms of floating point operations, or FLOPs) typically scales with the size of the mesh (*e.g.,* the number of nodes). The task is to evolve the system long-term into the future, with a low error and a constraint on the computational cost.

Most classical solvers use a fixed mesh $E^t$ whose topology does not vary with time. For example, the mesh $E^t \equiv E^0$ can be a 2D or 3D regular grid, or an irregular mesh that is pre-generated at the beginning of the simulation (Geuzaine & Remacle, 2009). Classical Adaptive Mesh Refinement (AMR) (Narain et al., 2012) addresses the multi-resolution challenge by adaptively refine or coarsen the mesh resolution, with heuristics based on local state variation. Since they are based on classical solvers, they may not benefit from the many advantages that deep learning brings (GPU acceleration, less stringent spatial and temporal resolution requirement, explicit forward, etc.). In contrast, our LAMP is a deep-learning based surrogate model, and can benefit from the many advantages (*e.g.,* speedup) offered by the deep learning framework. Furthermore, since it directly optimize for a linear combination of error and computation, it has the potential to directly optimize to a better error vs. computation tradeoff, nearer to the true Pareto frontier.

Deep-learning based surrogate models, although having achieved speedup compared to classical solvers, still typically operate on a fixed grid or mesh (Li et al., 2021; Sanchez et al., 2020; Wu et al., 2022; Zhao et al., 2022; Han et al., 2022), and have yet to exploit the multi-resolution nature typical in physical simulations. One important exception is MeshGraphNets (Pfaff et al., 2021), which both learns how to evolve the state $V^t$, and uses supervised learning to learn the spatial adaptation that changes $E^t$. However, since it uses supervised learning where the ground-truth mesh is provided from the classical solver with AMR, it cannot exceed the performance of AMR in terms of error vs. computation tradeoff, and has to interact with the classical solver in inference time to perform the adaptive mesh refinement. In contrast, our LAMP directly optimizes for the objective, which uses reinforcement learning for learning the policy of refinement and coarsening, and has the potential to surpass classical AMR and achieve a better error vs. computation tradeoff. Moreover, a single trained LAMP can adapt to the full range of relative importance $\beta$ of error vs. computation at inference time, thus can be more versatile than MeshGraphNets with a fixed strategy. Another pioneering work by Yang et al. (2021) learns the adaptive remeshing using RL. It has notable differences with our work. Firstly, their method is evaluated for the specific finite element method (FEM), and cannot be directly applied for more general simulations, *e.g.*, cloth simulations as in our experiment. Furthermore, our method is the first method that jointly learns the remeshing and evolution. Secondly, while the goal of Yang et al. (2021) is to reduce error, ours is to learn a controllable tradeoff between reducing error and reducing computational cost. Thirdly, the actions of Yang et al. (2021) are refinement on the *faces* of the rectangular meshes, while our actions are refinement and coarsening on the *edges* of triangular meshes. Fourthly, LAMP does not require the classical solver in the loop, thus significantly reducing the training time.

## 3 METHOD

In this section, we detail our LAMP method. We first introduce its architecture in Sec. 3.1. Then we introduce its learning method (Sec. 3.2), including learning objective and training, and technique to let it learn to adapt to varying importance of error and computation. The high-level schematic of our LAMP is shown in Fig. 1.

### 3.1 MODEL ARCHITECTURE

The model architecture of LAMP consists of two components: an actor-critic which updates the mesh topology, and an evolution model which evolves the states defined on the mesh. We will detail them one by one.

**Actor-critic**. The actor-critic consists of a policy network $f_\varphi^{\text{policy}}$ (with parameter $\varphi$) which predicts the probability of performing the spatial coarsening or refining actions, and a value network $f_\varphi^{\text{value}}$ which evaluates the long-term expected reward of the policy network:

$$P(A = a^t | M = M^t, \beta) = p_\varphi(a^t | M^t, \beta) = f_\varphi^{\text{policy}}(M^t, \beta) \tag{2}$$

$$\hat{v}^t = f_\varphi^{\text{value}}(M^t, \beta) \tag{3}$$

where $a^t$ is the (refining and coarsening) action performed on the edges $E^t$ so that it will become $\hat{E}^{t+1}$. The policy network $f_\varphi^{\text{policy}}$ outputs the probability of performing such action and can sample from this probability. $v^t$ estimates the "value" of the current policy starting from current state $M^t$ (for more information, see Sec. 3.2 below). The explicit dependence on $\beta$ (as the $\beta$ in Eq. 8) allows the policy and value network to condition on the varying importance of error and computation. Given the predicted mesh $\hat{E}^{t+1}$ and the current node features $V^t$ on the current mesh $E^t$, an interpolation $g^{\text{interp}}$ is performed which obtains the node features on the new mesh (see Appendix B for details):

$$\hat{V}'^t = g^{\text{interp}}(V^t, \hat{E}^{t+1}, E^t) \tag{4}$$

Now the new intermediate state $\hat{M}'^t = (\hat{V}'^t, \hat{E}^{t+1})$ is defined on the new mesh $\hat{E}^{t+1}$.

**Evolution model**. The second component is an evolution model $f_\theta^{\text{evo}}$ which takes as input the intermediate state $M'^t$ defined on $\hat{E}^{t+1}$, and outputs the prediction of node features $\hat{V}^{t+1}$ for time $t+1$:

$$\hat{V}^{t+1} = f_\theta^{\text{evo}}(M'^t) \tag{5}$$

Note that in this stage, the mesh topology $\hat{E}^{t+1}$ is kept constant, and the evolution model $f_\theta^{\text{evo}}$ (with parameter $\theta$) learns to predict the state based on the current mesh.

Taken together, Eqs. (2)(4)(5) evolve the system state from $M^t$ at time $t$ to state $\hat{M}^{t+1} = (\hat{V}^{t+1}, \hat{E}^{t+1})$ at $t+1$. During inference, they are executed autoregressively following Eq. (1), to predict the system's future states $\hat{M}^t, t = 1, 2, 3, ...$, given an initial state $M^0$.

**GNN architecture**. One requirement for the evolution model $f_\theta^{\text{evo}}$, policy network $f_\varphi^{\text{policy}}$ and value network $f_\varphi^{\text{value}}$ is that they can operate on changing mesh topology $E^t$. Graph Neural Networks (GNNs) are an ideal choice that meets this requirement. Since we represent the system's state as mesh, we adopt MeshGraphNets (Pfaff et al., 2021) as the base architecture for the above three models. Specifically, we encode $V^t$ as node features for the graph, and encode the mesh topology $E^t$ as edges and world edges as two types of edges, and the edge features depend on the relative positions in the mesh coordinates and world coordinates. Based on the graph, a processor network that consists of $N$ layers of message passing are performed to locally exchange and aggregate the information:

$$Z_{ij}^{(e)n+1} = \text{MLP}_\theta^{(v)}(E_{ij}^n, Z_i^{(v)n}, Z_j^{(v)n}) \tag{6}$$

$$Z_i^{(v)n+1} = \text{MLP}_\theta^{(v)}(Z_i^{(v)n}, \sum_j Z_{ij}^{(e)n+1}). \tag{7}$$

where $Z_i^{(v)n}$ is the latent node vector on node $i$ at layer $n$, and $Z_{ij}^{(e)n}$ is the latent edge vectors on the $n^{\text{th}}$ layer on the edge from node $i$ to node $j$. We have $Z_i^{(v)0} = \hat{V}_i'^t$ and $Z_i^{(v)N} = \hat{V}_i^{t+1}$ are input and predicted node features at time $t$ and $t+1$, respectively, in Eq. (5). Figure 1 provides an illustration of the architecture. We use an independent processor for the evolution model, and share the processor for the policy and value networks. After the processor, the latent vectors are concatenated with $\beta$ to feed into downstream decoders. For the evolution model $f_\theta^{\text{evo}}$, a decoder is operated on the latent state and outputs the prediction $\hat{V}'^{t+1}$ on the nodes. For the value network, an value MLP is operated on all nodes, and a global pooling is performed to compute the overall estimated value. For the policy network, we design the action decoder as follows.

**Action representation**. To predict the action for policy network and its probability, we first need to design the action space. Note that compared to typical reinforcement learning problems, here the action space is extremely high-dimensional and complicated: (1) each edge of the mesh may have the option of choosing refinement or coarsening. If there are thousands of edges $N_{\text{edge}}$, then the possible actions will be on the order of $2^{N_{\text{edge}}}$. (2) Not all actions are valid, and many combinations of actions are invalid. For example, two edges on the same face of the mesh cannot be refined at the same time, nor can they be both coarsened. To address this high-dimensionality action problem, we introduce the following design of action space, where for both refinement and coarsening, the policy network first samples integers $K^{\text{re}}, K^{\text{co}} \in \{0, 1, 2, ...K^{\text{max}}\}$, and then independently samples $K^{\text{re}}$ edges to perform refinement and $K^{\text{co}}$ edges to perform coarsening with proper filtering. The full sampled action is $a^t = (K^{\text{re}}, e_1^{\text{re}}, e_2^{\text{re}}, ...e_{K^{\text{refine}}}^{\text{re}}, K^{\text{co}}, e_1^{\text{co}}, e_2^{\text{co}}, ...e_{K^{\text{co}}}^{\text{co}})$, where $K^{\text{re}}, K^{\text{co}} \in \{0, 1, ...K^{\text{max}}\}$, and $e_k^{\text{re}}, e_k^{\text{co}} \in E^t, k = 1, 2, ...$ The log-probability for the sampled action $a^t$ is given by:

$$\log p_\varphi(a^t|M^t) = \log p_\varphi(K^{\text{re}}|M^t) + \sum_{k=1}^{K^{\text{re}}} \log p_\varphi(e_k^{\text{re}}|M^t) + \log p_\varphi(K^{\text{co}}|M^t) + \sum_{k=1}^{K^{\text{co}}} \log p_\varphi(e_k^{\text{co}}|M^t)$$

## 3.2 Learning

The ultimate goal of the learning for LAMP is to optimize the objective Eq. (8) as follows:

$$L = (1 - \beta) \cdot \text{Error} + \beta \cdot \text{Computation} \tag{8}$$

for a wide range of $\beta$. To achieve this, we first pre-train the evolution model without remeshing to obtain a reasonable evolution model, then break down the above objective into an alternative learning of two phases (Appendix B.1): learning the evolution model with objective $L^{\text{evo}}$ that minimizes long-term evolution error, and learning the policy with objective $L^{\text{policy}}$ that optimizes both the long-term evolution error and computational cost.

**Learning evolution**. In this phase, the evolution model $f_\theta^{\text{evo}}$ is optimized to reduce the multi-step evolution *error* for the evolution model. As before, we denote $M^{t+s}, t = 0, 1, 2, ..., s = 0, 1, ...S$ as the state of the system at time $t + s$ simulated by the ground-truth solver with very fine-grained mesh, and denote $\hat{M}^{t+s}, t = 0, 1, 2, ..., s = 0, 1, 2, ..S$ as the prediction by the current LAMP following the current policy, up to a horizon of $S$ steps into the future. We further denote $\hat{M}''^{t+s}, t = 0, 1, 2, ..., s = 0, 1, 2, ..S$ as the prediction by the current evolution model on the fine-grained mesh, where its mesh is provided as ground-truth mesh $E^{t+s}$ at each time step. Then the loss is given by:

$$L^{\text{evo}} = L_S^{\text{evo}}[f_\varphi^{\text{policy}}, f_\theta^{\text{evo}}; \hat{M}^t] + L_S^{\text{evo}}[\mathbb{I}, f_\theta^{\text{evo}}; \hat{M}''^t] \tag{9}$$

$$= \sum_{s=1}^{S} \alpha_s^{\text{policy}} \ell(\hat{M}^{l+s}, M^{l+s}) + \sum_{s=1}^{S} \alpha_s^{\mathbb{I}} \ell(\hat{M}''^{l+s}, M^{l+s}) \tag{10}$$

Essentially, we optimize two parts of the evolution loss: (1) $L_s^{\text{evo}}[f_\varphi^{\text{policy}}, f_\theta^{\text{evo}}; \hat{M}^t]$ which is the evolution loss by following policy network $f_\varphi^{\text{policy}}$ and evolution model $f_\theta^{\text{evo}}$, starting at initial state of $\hat{M}^t$ for $S$ steps. (here $\alpha_s^{\text{policy}}$ is the coefficient for the $s$-step loss with loss function $\ell$). This makes sure that evolution model $f_\theta^{\text{evo}}$ adapts to the current policy $f_\varphi^{\text{policy}}$ that designates proper computation. (2) The second part of the loss, $L_s^{\text{evo}}[\mathbb{I}, f_\theta^{\text{evo}}; \hat{M}''^t]$, is the evolution loss by using the ground-truth mesh and evolved by the evolution model $f_\theta^{\text{evo}}$, starting at initial state of fine-grained mesh $\hat{M}''^t$ and evolve for $s$ steps. This encourages the evolution model to learn to utilize more computation to achieve a better prediction error, if the mesh is provided by the ground-truth mesh.

**Learning the policy**. In this phase, the policy network $f_\varphi^{\text{policy}}$ learns to update the spatial resolution (refinement or coarsening of mesh) at each location, to improve both the *computation* and the prediction *error*. Since the spatial refinement and coarsening are both discrete action, and the metric of computation is typically non-differentiable, we use Reinforcement Learning (RL) to learn the policy. Specifically, we model it as a Markov Decision Process (MDP), where the environment state is the system's state $M^t$, the actions are the local refinement or coarsening at each edge of $E^t$, and we design the reward as the *improvement* on both the error and computation, between following the current policy's action, and an *counterfactual* scenario where the agent follows an identity policy $\mathbb{I}$

that does not update the mesh topology, starting on the initial state $\hat{M}^t$. Concretely, the reward is

$$r^t = (1 - \beta) \cdot \Delta\text{Error} + \beta \cdot \Delta\text{Computation} \tag{11}$$

$$\Delta\text{Error} = L_S^{\text{evo}}[\mathbb{I}, f_\theta^{\text{evo}}; \hat{M}^t] - L_S^{\text{evo}}[f_\varphi^{\text{policy}}, f_\theta^{\text{evo}}; \hat{M}^t] \tag{12}$$

$$\Delta\text{Computation} = \mathcal{C}_S[\mathbb{I}, f_\theta^{\text{evo}}; \hat{M}^t] - \mathcal{C}_S[f_\varphi^{\text{policy}}, f_\theta^{\text{evo}}; \hat{M}^t] \tag{13}$$

Here $\mathcal{C}_S[\cdot]$ is a surrogate metric that quantifies "Computation" based on the predicted mesh topology $\hat{E}^{t+1}, \hat{E}^{t+2}, ...\hat{E}^{t+S}$ up to $S$ steps into the future. In this paper we use the number of nodes as the surrogate metric for measuring the computation, since typically for the GNNs, the computation (in terms of FLOPs) scales linearly with the number of nodes (since each node has a bounded number of edges on the mesh, the number of edges thus scales linearly with number of nodes, so will message passing and node updates).

To optimize the reward $r^t$, we employ the standard REINFORCE as used in (Sutton et al., 1999; Hafner et al., 2021) to update the policy network $f_\varphi^{\text{policy}}$, with the following objective:

$$L_\beta^{\text{actor}} = \mathbb{E}_t \left[ -\log p_\varphi(a^t|M^t, \beta)\text{sg}(r^t - f_\varphi^{\text{value}}(M^t, \beta)) - \eta \cdot \text{H}[p_\varphi(a^t|M^t, \beta)] \right] \tag{14}$$

Here $\text{H}[\cdot]$ is the entropy, which encourages the action to have higher entropy to increase exploration, where $\eta$ here is a hyperparameter. The $\text{sg}(\cdot)$ is stop-gradient. Essentially, the first term in loss $L^{\text{policy}}$ encourages to increase the log-probability of actions that have a higher "advantage", where the advantage is defined as the difference between the current reward $r^t$ that follows the current action $a^t$ taken, and the expected reward (value) $f_\varphi^{\text{value}}(M^t, \beta)$ that follows the current policy starting from the current state $M^t$. We can also think of it as an actor-critic where the critic tries to evaluate accurately the expected reward of the current policy, and an actor (policy) is trying to exceed that expectation. To train the value network, we use MSE loss:

$$L_\beta^{\text{value}} = \mathbb{E}_t \left[ (f_\varphi^{\text{value}}(M^t, \beta) - r^t)^2 \right] \tag{15}$$

**Learning to adapt to varying $\beta$.** In the overall objective (Eq. 8), the $\beta$ stipulates the relative importance between Error and Computation. $\beta = 0$ means we only focus on minimizing Error, without constraint on Computation. $\beta = 1$ means we only focus on minimizing computation, without considering the evolution error. In practice, we typically wish to improve both, with a $\beta \in (0, 1)$ that puts more emphasis on one metric but still considers the other metric. To allow LAMP to be able to operate at varying $\beta$ at inference time, during the learning of policy, we sample $\beta$ uniformly within a range $\mathcal{B} \subseteq [0, 1]$ (*e.g.*, $\mathcal{B}$ can be $[0, 1]$ or $[0, 0.5]$), for different examples within a minibatch, and also train the policy and value network jointly, where the total loss $L^{\text{policy}}$ is the weighted sum of the two policy and value losses:

$$L^{\text{policy}} = \mathbb{E}_{\beta \sim \mathcal{B}}[L_\beta^{\text{actor}} + \alpha^{\text{value}} \cdot L_\beta^{\text{value}}] \tag{16}$$

where $\alpha^{\text{value}}$ is a hyperparameter, which we set as 0.5. In this way, the policy can learn a generic way of spatial coarsening and refinement, conditioned on $\beta$. For example, for smaller $\beta$ that focuses more on improving error, the policy network may learn to refine more on dynamic regions and coarsen less, sacrificing computation to improve prediction error.

## 4 EXPERIMENTS

In the experiments, we set out to answer the following questions on our proposed LAMP:

- Can LAMP learn to coarsen and refine the mesh, focusing more computation on the more dynamic regions to improve prediction accuracy?
- Can LAMP improve the Pareto frontier of Error vs. Computation, compared to state-of-the-art deep learning surrogate models?
- Can LAMP learn to condition on the $\beta$ to change is behavior, and perform varying amount of refinement and coarsening depending on the $\beta$?

We evaluate our LAMP on two challenging datasets: (1) a 1D benchmark nonlinear PDEs , which tests generalization of PDEs in the same family (Brandstetter et al., 2022); (2) a mesh-based paper simulation generated by the ArcSim solver (Narain et al., 2012). Both datasets possess multi-resolution characteristics where some parts of the system is highly dynamic, while other parts are changing more slowly.

## 4.1 1D NONLINEAR FAMILY OF PDEs

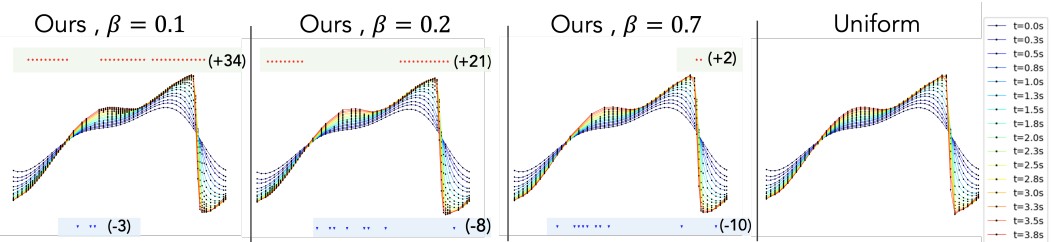

Figure 2: Example rollout result of our LAMP on 1D nonlinear PDEs. The rollout is performed over 200 time steps, where different color denotes the system's state at different time. On each state, we also plot the corresponding mesh as black dots. The upper green and lower blue band shows the added and removed nodes of the mesh, comparing the end mesh and initial mesh. We see that with a smaller $\beta$ (*e.g.*, $\beta = 0.1$) that emphasizes more on "Error", it refines more on highly-dynamic regions (near shock front) and coarsens less. With a larger $\beta$ (*e.g.*, $\beta = 0.7$) that focuses more on reducing computation, it almost doesn't refine, and choose to coarsen on more static regions.

**Data and Experiments**. In this section, we test LAMP's ability to balance error vs. computation tested on unseen equations with different parameters in a given family. We use the 1D benchmark in Brandstetter et al. (2022), whose PDEs are given by

$$\left[\partial_t u + \partial_x(\alpha u^2 - \beta \partial_x u + \gamma \partial_{xx} u)\right](t, x) = \delta(t, x) \tag{17}$$

$$u(0, x) = \delta(0, x), \quad \delta(t, x) = \sum_{j=1}^{J} A_j \sin(\omega_j t + 2\pi \ell_j x / L + \phi_j) \tag{18}$$

The parameter for the PDE is given by $p = (\alpha, \beta, \gamma)$. The term $\delta$ is a forcing term (Bar-Sinai et al., 2019) with $J = 5, L = 16$ and coefficients $A_j$ and $\omega_j$ sampled uniformly from $A_j \sim U[-0.5, 0.5]$, $\omega_j \sim U[-0.4, 0.4]$, $\ell_j \in \{1, 2, 3\}$, $\phi_j \sim U[0, 2\pi)$. We uniformly discretize the space to $n_x = 200$ in $[0, 16)$ and uniformly discretize time to $n_t = 250$ points in $[0, 4]$. Space and time are further downsampled to resolutions of $(n_t, n_x) \in \{(250, 100), (250, 50), (250, 25)\}$ as initial resolution. We use the **E2** scenario in the benchmark, which tests the model's ability to generalize to novel parameters of PDE with the same family. Specifically, we have that the parameter $p = (1, \eta, 0)$ where $\eta \sim U[0, 0.2]$.

As our LAMP autoregressively simulate the system, it can refine or coarsen the mesh at appropriate locations by the policy network $f_\varphi^{\text{policy}}$, before evolving to the next state with the evolution model $f_\theta^{\text{evo}}$. We evaluate the models with the metric of Computation and long-term evolution Error. For the computation, we use the average number of vertices throughout the full trajectory as a surrogate metric, since the number of floating point operations typically scales linearly with the number of vertices in the mesh. For long-term evolution error, we use the cumulative MSE over 200 steps of rollout, starting with initial state from time steps 25 to 49. We compare LAMP with strong baselines of deep learning-based surrogate models, including CNNs, Fourier Neural Operators (FNO) (Li et al., 2021), and MP-PDE (Brandstetter et al., 2022) which is a state-of-the-art deep learning-based surrogate models for this task. Our base neural architecture is based on MeshGraphNets (Pfaff et al., 2021) which is a state-of-the-art GNN-based model for mesh-based simulations. We compare an ablation of our model that does not perform remeshing (LAMP no remeshing), and a full version of our model. For all models, we autoregressively roll out to predict the states for a full trajectory length of 200, using the first 25 steps as initial steps. We perform three groups of experiments, starting at initial number vertices of 25, 50 and 100 that is downsampled from the 100-vertice mesh, respectively. The three baselines all do not perform remeshing, and our full model has the ability to perform remeshing that coarsen or refine the edges at each time step. We record the accumulated MSE as measure for error, and average number of vertices over the full rollout trajectory as metric for computational cost. Note that for all models, the MSE is computed on the full ground-truth mesh with 100 vertices, where the values of prediction are linearly interpolated onto the location of the ground-truth. This prevents the model to "cheat" by reducing the number of vertices and only predict well on those vertices. Additional details of the experiments are given in Appendix B.2.

Table 1: Result (error and computational cost) for 1D family of nonlinear PDEs, for a total rollout trajectory of 200 time steps, providing the first 25 steps as input. LAMP significantly reduces the long-term prediction error compared to baselines, especially for smaller number of initial vertices (error reduction of 71.5% for 25 initial vertices, and 23.5% for 50 initial vertices), when evaluating on ground-truth mesh that has 100 vertices. This shows that LAMP is able to improve prediction error by selecting where to focus computation on, especially with more stringent computational constraint. Note that we limit the maximum #vertices to be 100 for all models.

| Model | Initial # vertices | Average # vertices | Error (MSE) |
|---|---|---|---|
| CNN | 25 | 25.0 | 4.75 |
| FNO | 25 | 25.0 | 4.85 |
| MP-PDE | 25 | 25.0 | 3.69 |
| LAMP (no remeshing) | 25 | 25.0 | 6.39 |
| **LAMP (ours)** | 25 | 37.6 | **1.05** |
| CNN | 50 | 50.0 | 1.10 |
| FNO | 50 | 50.0 | 1.79 |
| MP-PDE | 50 | 50.0 | 0.98 |
| LAMP (no remeshing) | 50 | 50.0 | 2.15 |
| **LAMP (ours)** | 50 | 53.2 | **0.75** |
| CNN | 100 | 100.0 | 0.81 |
| FNO | 100 | 100.0 | 1.39 |
| MP-PDE | 100 | 100.0 | 0.88 |
| LAMP (no remeshing) | 100 | 100.0 | **0.75** |
| **LAMP (ours)** | 100 | 100.0 | 0.76 |

**Results**.   Table 1 shows the results.   We see that our LAMP outperforms all baselines by a large margin, achieving an error reduction of 71.5%, 23.5% and 6.2% (average of 33.7%) on initial #nodes of 25, 50 and 100, respectively, compared with the best performing baseline of CNN, FNO and MP-PDE. Importantly, we see that compared with an ablation with no remeshing, our full LAMP is able to significantly reduce error (by 83.6% error reduction for 25 initial vertices scenario, and 65.1% for 50 vertices scenario), with only modest increase of average number of vertices (by 50.4% and 6.4% increase, respectively). This shows the ability of LAMP to adaptively trade computation to improve long-term prediction error.

To investigate whether LAMP is able to focus computation on the most dynamic region, we visualize example trajectory of LAMP, as shown in Fig. 2 and Fig. 8 in Appendix C.1. Starting with 50 vertices, we test our model on different $\beta$, where smaller $\beta$ focuses more on improving error. We see that with smaller $\beta$ (*e.g.*, $\beta = 0.1$), LAMP is able to add more nodes (25 nodes) on the most dynamic region, and only coarsen few (removing 3 nodes in total). With a larger $\beta$ that focuses more on reducing computation, we see that LAMP refines less and coarsen more, and only coarsen on the more slowly changing region. Additionally, we visualize the error (y-axis) vs. number of vertices for varying $\beta$ for two different models over all test trajectories in Fig. 3. We see that with increasing $\beta$, LAMP is able to reduce the vertices more, with only slight increase of error. Furthermore, LAMP significantly improves the Pareto frontier. In summary, the above results show that LAMP is able to focus computation on dynamic regions, and able to adapt to different $\beta$ at inference time.

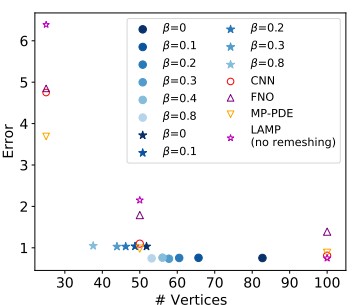

Figure 3: Error (MSE) vs. average number of vertices with varying $\beta$, for two LAMP models trained with initial number of vertices of 25 (marked by solid stars) and 50 (solid circles), and baselines (hollow markers). We see LAMP improves the Pareto frontier.

### 4.2   2D MESH-BASED SIMULATION

Here we evaluate our LAMP's ability on a more challenging setting with paper folding simulation. The paper in this simulation is square-shaped and its boundary edge length is 1. During the simula-

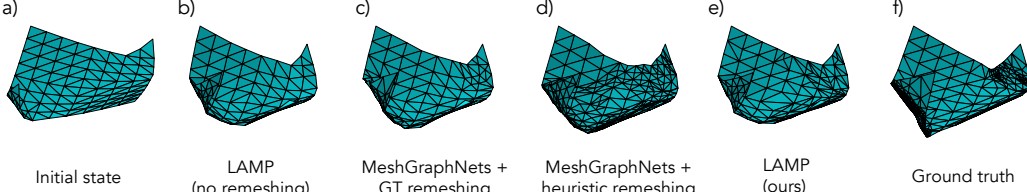

Figure 4: Example result of 2D mesh-based paper simulation. We observed that LAMP is adding more resolution to the high-curvature region to resolve the details and coarsen the middle flat region. Figure a) is at $t = 0$, and figure b), c), d), e), f) are LAMP (no remeshing), MeshGraphNets with ground-truth mesh, MeshGraphNets with heuristic remeshing, LAMP (ours), and the ground-truth results at $t = 20$. Additional visualization could be found in Figure 9.

Table 2: Computation vs. Error for 2D mesh-based paper simulation for different methods. With the proposed learned remeshing framework, LAMP are able to achieve better roll-out error with slight increase of average number of vertices. Reported number is the MSE over 20 learned-simulator-steps roll-out, starting at initial states at steps 10, 30, and 50, averaged over 50 test trajectories.

| Model | Initial # vertices | Average # vertices | Error (MSE) |
|---|---|---|---|
| MeshGraphNets + GT remeshing | 102.9 | 115.9 | 5.91e-4 |
| MeshGraphNets + heuristics remeshing | 102.9 | 191.9 | 6.38e-4 |
| LAMP (no remeshing) | 102.9 | 102.9 | 6.13e-4 |
| **LAMP (ours)** | 102.9 | 123.1 | **5.80e-4** |

tion, 4 corners of the paper will receive different magnitude of force. When generating ground-truth trajectories by the ArcSim solver (Narain et al., 2012), we set the minimum and maximal length of edges to be 0.01 and 0.2. We evaluate the models with the metric of Computation and long-term evolution error. Similar to the Section 4.1, we use the average number of nodes throughout the full trajectory as a surrogate metric for complexity. We also compare with two baselines. The first baseline is MeshGraphNets with ground-truth (GT) remeshing, where the remeshing is provided by the ArcSim's Adaptive Anisotropic Remeshing component. This is used in (Pfaff et al., 2021) and provides lower error than learned remeshing. The second baseline is MeshGraphNets + heuristics remeshing, where the heuristics refines the edge based on the local curvature (Appendix B.4).

As shown in Table 2, our model is able to add more resolution to the high curvature region, and achieve better roll-out accuracy than the ablation without remeshing and baselines. we see that our LAMP outperforms both baselines and the no-remeshing ablation. Specifically, LAMP outperforms the strong baseline of "MeshGraphNets + GT remeshing". This shows that LAMP can further improve upon MeshGraphNets with ground-truth remeshing to learn a better remeshing policy, allowing the evolution model to evolve the system in a more faithful way. Furthermore, the "Mesh-GraphNets + heuristic remeshing" baseline has a larger error, showing that this intuitive baseline is suboptimal. Finally, LAMP outperforms its ablation without remeshing, showing the necessity of remeshing which can significantly reduce the prediction error. Additional details of the experiments are given in Appendix B.3. In Fig. 4 and Fig. 9 in Appendix C.2, we see that our LAMP learns to add more mesh onto the more dynamic regions near to the folding part (with high curvature), showing LAMP's ability to assign computation to the most needed region.

## 5 CONCLUSION

In this work, we have introduced LAMP, the first fully deep learning-based surrogate model that jointly learns the evolution of physical system and optimizes assigning the computation to the most dynamic regions. In 1D and 2D datasets, we show that our method is able to adaptively perform refinement or coarsening actions, which improves long-term prediction error than strong baselines of deep learning-based surrogate models. We hope our method provides a useful tool for more efficient and accurate simulation of physical systems.

## 6 ACKNOWLEDGEMENT

We thank Ian Fischer and Xiang Fu for discussions and for providing feedback on our manuscript. We also gratefully acknowledge the support of DARPA under Nos. HR00112190039 (TAMI), N660011924033 (MCS); ARO under Nos. W911NF-16-1-0342 (MURI), W911NF-16-1-0171 (DURIP); NSF under Nos. OAC-1835598 (CINES), OAC-1934578 (HDR), CCF-1918940 (Expeditions), NIH under No. 3U54HG010426-04S1 (HuBMAP), Stanford Data Science Initiative, Wu Tsai Neurosciences Institute, Amazon, Docomo, GSK, Hitachi, Intel, JPMorgan Chase, Juniper Networks, KDDI, NEC, and Toshiba.

The content is solely the responsibility of the authors and does not necessarily represent the official views of the funding entities.

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

# Appendix

## A  MODEL ARCHITECTURE

Here we detail the architecture of LAMP, complementary to Sec. 3.1. This architecture is used throughout all experiment, with just a few hyperparameter (*e.g.*, latent dimension, message passing steps) depending on the dimension (1D, 2D) of the problem.

### A.1  ARCHITECTURE

In this subsection, we first detail the base architecture that is common among the evolution model $f_\theta^{\text{evo}}$, the policy network $f_\varphi^{\text{policy}}$, and value network $f_\varphi^{\text{value}}$. Then describe their respective aspects, and explain the benefits of our action representation.

**Base architecture**. We use MeshGraphNets Pfaff et al. (2021) as our base architecture, for the evolution model $f_\theta^{\text{evo}}$, the policy network $f_\varphi^{\text{policy}}$, and value network $f_\varphi^{\text{value}}$. All three models have an encoder and a processor. For the encoder, we encode both node features and edge features, uplift to latent dimension using MLPs, to $Z_{ij}^e$, $Z_i^v$. For the processor, it consists of $N$ message passing layers. For each message-passing layer, we first update the edge features using current edge features, and connected node features, $Z_{ij}^{(e)n+1} = \text{MLP}_\theta^{(v)}(E_{ij}^n, Z_i^{(v)n}, Z_j^{(v)n})$, then we update the node features using current node feature and connected edge features, $Z_i^{(v)n+1} = \text{MLP}_\theta^{(v)}(Z_i^{(v)n}, \sum_j Z_{ij}^{(e)n+1})$. Depending on the models and their intended functions, different models have different modules for predicting the output, described as follows.

**Evolution model**. For the evolution model, the output is at each node, and uses the last latent vector on the node, feeds into a decoder, to predict the output $\hat{V}_i^{t+1} = Z_i^{(v)N}$ (See Eq. 6) at the next time step $t + 1$.

**Policy network and value network**. Both $f_\varphi^{\text{policy}}$ and $f_\varphi^{\text{value}}$ shares the same processor $f_\varphi^{\text{processor}}$, which has $N$ message-passing layers. The final node feature is appended with $\beta$ for the controllability. The $f_\varphi^{\text{policy}}$ takes the encoded graph as input and consist of two parts - a global mean pooling is used to obtain the global latent representation of the whole graph, which is then feed into a $\text{MLP}^{(k)}$ followed by a action specific linear layer to predict the probability distribution for different number of actions should be performed; an action specific $\text{MLP}^{(a)}$ is applied on each edge to predict the probability of a certain action (*i.e.*, split, coarse) to be applied on this edge. The $f_\varphi^{\text{value}}$ takes the same encoded graph as input, and perform a global mean pooling to obtain the global latent features for the graph, which is then feed through a linear layer to predict the accuracy reward, and another linear layer to predict the computation reward, the final predicted value is, $v = \text{loss} + \beta \cdot \text{compute}$.

### A.2  ARCHITECTURAL HYPERPARAMETERS USED IN 1D AND 2D EXPERIMENTS

Here we detail the architectures used in the 1D and 2D experiment. A summary of the hyperparameters is also provided in Table 3.

#### A.2.1  1D NONLINEAR PDES

For 1D experiment, our evolution model $f_\theta^{\text{evo}}$ has $N = 3$ message-passing layers in the processor and latent dimension of 64. It uses the SiLU activation (Elfwing et al., 2018). The shared processor for $f_\varphi^{\text{policy}}$ and $f_\varphi^{\text{value}}$ has $N = 3$ message-passing layers and latent dimension of 64.

#### A.2.2  2D MESH-BASED SIMULATION

For $f_\theta^{\text{evo}}$, we set message passing layers for MeshGraphNets (Pfaff et al., 2021) to be 8. We also model our message passing function with MLP having 56 units followed by SiLU activation (Elfwing et al., 2018). Here, we share the same message passing function across all the 8 layers because we found in our experiments that sharing the same message passing function achieved better

performance than having independent MLP for each of the layers. The shared processor for $f_\varphi^{\text{policy}}$ and $f_\varphi^{\text{value}}$ has $N = 3$ message-passing layers and latent dimension of 64.

# B  EXPERIMENT DETAILS

In this section, we provide experiment details for 1D and 2D datasets. Firstly, we explain the reasoning behind several design choices of LAMP and their benefits. Then we provide the training procedure summary and hyperparameter table in Appendix B.1, followed by specific training details for 1D (Appendix B.2) and 2D (Appendix B.3). Finally, we detail the baselines in Appendix B.4.

**Use of $r^t$ as value target**. Different from typical RL learning scenario (computer games or robotics) where the episode always has an end and the reward is bounded, here in physical simulations, there are two distinct characteristics as follows. (1) The rollout can be performed infinite time steps into the future, (2) as the rollout continues, at some point the error between predicted state and ground-truth state will diverge, so the error is not bounded. Based on these two characteristics, a value target based on the error for infinite horizon (*e.g.*, the one used in Dreamer v2 (Hafner et al., 2021)) does not make sense, since the error will not be bounded. In our experiment, we also observe similar phenomena, in which both the value target and value prediction continue to increase indefinitely. Thus, we use the average reward in the $S$ step rollout as the value target, which measures the error and computation improvement within the rollout window we care about and proves to be much more stable.

**Benefit of our action space definition**. we have provided the description of action representation in Section our definition of action space in Section 3.1. Compared to an independent sampling on each edge, the above design of action space has the following benefits:

- The action space reduces from $2^{N_{\text{edge}}}$ to $N_{\text{edge}}{}^{K^{\text{max}}}$, where $N_{\text{edge}}$ is the number of edges. In the case of $N_{\text{edge}} \sim 1000$ and $K^{\text{max}} \sim 10 \ll N_{\text{edge}}$, the difference in action space dimensionality is significant, *e.g.*, $2^{1000} = 10^{300}$ vs. $1000^{10} = 10^{30}$. Therefore, it is easier to credit assign the reward to appropriate action, with a smaller space of action.

- Compared with each edge performing action independently, now only $K \leq K^{\text{max}}$ actions of refinement or coarsening can be performed. Therefore, it will need to focus on a few actions that can best improve the objective.

- The sampling of $K$ will also make the policy more "controllable", since now the $K$ is explicitly dependent on the $\beta$, and learning how many refinement or coarsening action to take depending on $\beta$ is much easier than figuring out which concrete independent actions to take.

## B.1  TRAINING PROCEDURE SUMMARY

Here we provide the training procedure for learning the evolution model and the policy. There are two stages of training. The first stage is pre-training the evolution model alone without remeshing, and the second stage is alternatively learning the policy with RL and finetuning the evolution model. The detailed hyperparameter table for 1D and 2D experiments is provided in Table 3. We train all our models on an NVIDIA A100 80GB GPU.

**Pre-training**. In the pre-training stage, the evolution model is trained without remeshing, and the loss is computed by rolling out the evolution model for $S$ steps, and the loss is given by loss = (1-step loss) $\times$ 1 + (2-step loss) $\times$ 0.1 + ... + ($S$-step loss) $\times$ 0.1 (the number $S$ is provided in Table 3). We use a smaller weight for later steps, so that the training is more stable (since at the beginning of training, the evolution model can have large error, having too much weight on later steps could result in large error and make the training less stable). The pre-training lasts for certain number of epochs (see Table 3), before proceeding to the next stage.

**Joint training of policy and evolution model**. In this stage, the actor-critic and the evolution model are trained in an alternative fashion. Specifically, in the policy-learning phase, the evolution model is frozen, and the actor-critic is learned via reinforcement learning (see "learning the policy" part of Sec. 3.2) for $J^{\text{policy}}$ steps, and in the evolution-learning phase, the actor-critic is frozen, and the

evolution model is learned according to the "learning evolution" part of Sec. 3.2, for $J^{\text{evo}}$ steps. These two phases proceed alternatively.

The reasoning behind using alternating training strategy for actor-critic and evolution is as follows. When learning the actor-critic, the *return* for the RL is the expected reward based on the *current* policy and evolution model. If at the same time the evolution model is also optimized together, then the reward function will always be changing, which will likely make learning the policy harder. Therefore, we adopt the alternating training strategy, which is also widely used in many other applications, such as in GAN training.

In the following two subsections, we detail the action space and specific settings for 1D and 2D.

Table 3: Hyperparameters used for model architecture and training.

| Hyperparameter name | 1D dataset | 2D dataset |
|---|---|---|
| *Hyperparameters for model architecture:* | | |
| Temporal bundling steps | 25 | 1 |
| $f_\theta^{\text{evo}}$: Latent size | 64 | 56 |
| $f_\theta^{\text{evo}}$: Activation function | SiLU | SiLU |
| $f_\theta^{\text{evo}}$: Encoder MLP number of layers | 3 | 4 |
| $f_\theta^{\text{evo}}$: Processor number of message-passing layers | 3 | 8 |
| $f_\varphi^{\text{policy}}$: Latent size | 64 | 56 |
| $f_\varphi^{\text{policy}}$: Activation function | ELU | ELU |
| $f_\varphi^{\text{policy}}$: Encoder MLP number of layers | 2 | 2 |
| $f_\varphi^{\text{policy}}$: Processor number of message-passing layers | 3 | 3 |
| $f_\varphi^{\text{policy}}$: MLP$^{(k)}$: MLP number of layers | 2 | 2 |
| $f_\varphi^{\text{policy}}$: MLP$^{(k)}$: Activation function | ELU | ELU |
| $f_\varphi^{\text{policy}}$: MLP$^{(a)}$: MLP number of layers | 3 | 3 |
| $f_\varphi^{\text{policy}}$: MLP$^{(a)}$: Activation function | ELU | ELU |
| *Hyperparameters for training:* | | |
| $\beta$ sampling range $\mathcal{B}$ | $[0, 0.5]$ | $\{0\}$ |
| Loss function | MSE | L2 |
| $\alpha_s^{\text{policy}}$, for $s = 1, 2, ...$ (Eq. 10) | $\{1,1,1,...\}$ | $\{1,1,1,...\}$ |
| $\alpha_s^{\mathbb{I}}$, for $s = 1, 2, ...$ (Eq. 10) | $\{1,1,1,...\}$ | $\{1,1,1,...\}$ |
| Number of epochs for pre-training evolution model | 50 | 100 |
| Number of rollout steps $S$ to for multi-step loss during pre-training | 4 | 1 |
| Number of epochs for joint training of actor-critic and evolution model | 30 | 30 |
| $J^{\text{policy}}$: # of steps for updating the actor-critic during joint training | 200 | 200 |
| $J^{\text{evo}}$: # of steps for updating the evolution model during joint training | 100 | 100 |
| Batch size | 128 | 64 |
| Evolution model learning rate for pre-training | $10^{-3}$ | $10^{-3}$ |
| Evolution model learning rate during policy learning | $10^{-4}$ | $10^{-4}$ |
| Value network learning rate | $10^{-4}$ | $10^{-4}$ |
| Policy network learning rate | $5 \times 10^{-4}$ | $5 \times 10^{-4}$ |
| Optimizer | Adam | Adam |
| Coefficient for value loss | 0.5 | 0.5 |
| $K^{\text{max}}$: Maximum number of actions for coarsen or refine | 20 | 20 |
| Maximum gradient norm | 2 | 20 |
| Optimizer scheduler | cosine | cosine |
| Input noise amplitude | 0 | $10^{-2}$ |
| $S$: Horizon | 4 | 6 |
| Weight decay | 0 | 0 |
| $\eta$: Entropy coefficient | $10^{-2}$ | $2 \times 10^{-2}$ |

## B.2   1D NONLINEAR PDEs

The action space for 1D problem is composed of split and coarsen actions. The coarse action is initially defined on all edges, which is then sampled based on the predicted number of coarsening actions, and probability of each edge to be coarsened. Among those sampled edges, if two edges share a common vertex, only the rightmost one will be coarsened.

During pre-training, to let the evolution model adapt to the varying size of the mesh, we perform random vertex dropout, where we randomly sample 10% of the minibatch to perform dropout, and if a minibatch is selected for node dropout, for each example, randomly drop 0-30% of the nodes in the mesh.

Here the interpolation $g^{\mathrm{interp}}$ (in Eq. 4) uses barycentric interpolation (Berrut & Trefethen, 2004), where during refinement, the value of the node feature for the newly added node is a linear combination of its two neighbors' node features, depending on the coordinates of the newly-added node and the neighboring nodes.

## B.3   2D MESH-BASED SIMULATION

In this subsection, we provide details on definition of remeshing actions, invalid remeshing action, how to generate 2d mesh data based on Narain et al. (2012) as well as how to pre-train the evolution model on the generated data, and the interpolation method to remedy inconsistency between vertex configurations of different meshes.

**Action space**. There are three kinds of remeshing operations defined in the 2D mesh simulation: split, flip, and coarsen. The action space for RL is defined as the product of sets of splitting and coarsening operations (the flipping is automatically performed). Elements of each set indicate edges in a mesh and the policy function chooses up to $K^{\max}$ of the elements as edges to split or coarsen. The reason why the flip action is not an action of RL is because we flip all edges satisfying some condition (will be explained in the following). The rest of this paragraph provides details on respective remeshing actions. Split action in 2D case can be performed on any edges that are shared with two triangular faces in a mesh and also on the boundary edges of the mesh. The split action results in 4 triangles (Fig. 5a). Flip operation is also performed on edges shared with two faces, but it also requires some additional condition, that is when sum of angles at vertices located at opposite side of edges is greater than $\pi$: see also Fig 5b. In our remeshing function, all edges satisfying the condition are flipped, which is the reason why flip operation is not a component of the action space for RL. Finally, coarsening action has relatively strong conditions. One condition is that one of the source or target nodes of a coarsened edge needs to be of degree 4 and another condition is that all the faces connected to the node of degree 4 needs to have acute angles except angles around the node. See also Fig. 5c. We filter the sampled actions from $f_\varphi^{\mathrm{policy}}$ based on aforementioned conditions to get sets of valid edges to split and coarse.

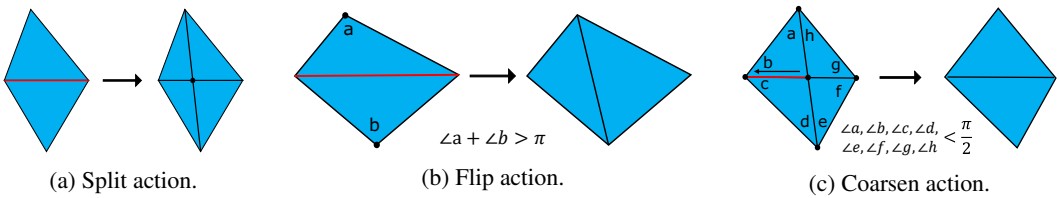

(a) Split action.    (b) Flip action.    (c) Coarsen action.

$\angle a + \angle b > \pi$

$\begin{aligned}\angle a, \angle b, \angle c, \angle d, \\ \angle e, \angle f, \angle g, \angle h\end{aligned} < \frac{\pi}{2}$

Figure 5: Illustration of split, flip and coarsen actions. The actions are performed on edges.

**Invalid remeshing**. As described in Section 3.1, two edges on the same face of a mesh cannot be refined at the same time, nor can they be both coarsened. This is because by doing so we may have an invalid mesh with some non-triangular faces such as quadrilaterals. We can give such an example as follows; see also Fig. 6 for reference. Suppose that we have a triangular face ABC and we split edges AB and AC. When we denote the midpoints of AB and AC by D and E, we have new edges DC and EB. If we denote the intersection of DC and EB by F, we will have a quadrilateral ADFE, which violates the requirement that all faces must be triangles. In order to avoid this situation, remeshing and coarsening action can only be performed on up to one edge of every face.

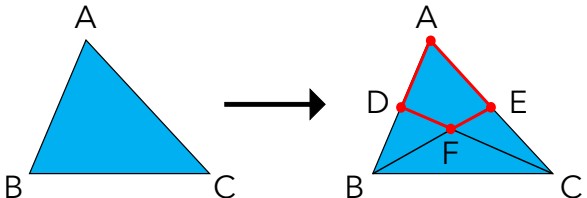

Figure 6: Invalid split action on edges. Splitting two edges in a same triangular face at the same time gives a quadrilateral.

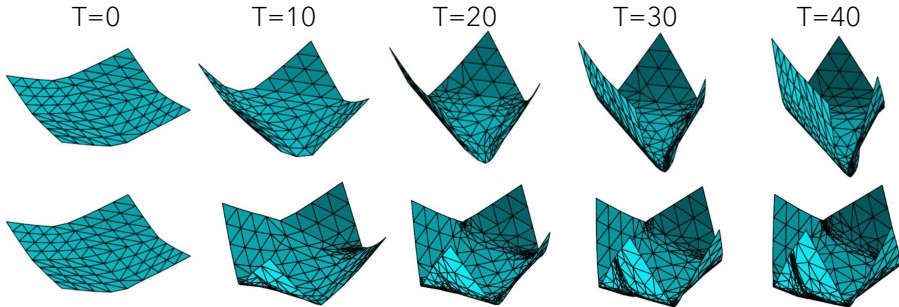

Figure 7: Part of trajectories generated with different configurations. As the time goes, finer triangular faces are added to high-curvature regions in meshes.

**Generating data**. To generate ground-truth data, we use a 2D triangular mesh-based AMR simulator Narain et al. (2012). This simulator adopts adaptive anisotropic kernel method to automatically conform to the geometric and dynamic detail of the simulated cloth: see also Fig. 7. In our experiment, each trajectory consists of 325 meshes (excluding initial mesh.) Each mesh consists of vertices and faces and every vertex is equipped with its 2D coordinates as well as 3D world coordinates. Time frame between two consecutive meshes are set to be 0.04. The edge length was set to range between 0.01 and 0.2. Note that decision on respective remeshing actions is made based on the 2D coordinates.

For training and test dataset, we generate 1050 configurations specifying direction and magnitude of forces applied to 4 corners of meshes, and generate 1000 trajectories for training data and use 50 trajectories as test data based on the generated configurations. When training our evolution model, we downsample data by half of the original data; we use every other meshes starting from an initial mesh in each trajectory. The reason of downsampling is that we observed that cumulative RMSE over rollout with model trained with full trajectories blew up after iteration exceeded 160 steps.

Noted that all dataset are pre-generated and will be loaded during the training, so that no ground-truth solver is needed during the training of LAMP.

**Pre-training of evolution model**. We model our evolution function $f_\theta^{\text{evo}}$ with MeshGraphNets (Pfaff et al., 2021). The input of the model adopts a graph representation where a vertex is equipped with 3D velocity computed from mesh information at both current and past time steps, and feature for an edge consists of relative distance and its magnitude of boundary vertices of the edge. Since mesh topology varies during the forward iteration, we perform barycentric interpolation on the mesh at past time step to get interpolated vertex coordinates corresponding to vertices at current time step.

When evaluating our model, we use rollout RMSE, taking the mean for all spatial coordinates, all mesh nodes, all steps in each trajectory, and all 50 trajectories in the test dataset. The rollout RMSE achieved $4.45 \times 10^{-2}$ with our best parameter setting. We found that adding noise helped to improve its performance; we added the noise with scale 0.01 to vertex coordinates.

**Interpolation**. Since we may have different mesh topology at each time step in a trajectory, it is not possible to simply compare node features at one time step with those at another time steps or even at the same time step if the mesh to compare is in a different trajectory. When we compare node features on different meshes, we perform barycentric interpolation (Berrut & Trefethen, 2004). For velocity in 2D simulation used as the input of the evolution model, we interpolate mesh at time step

$t - 1$ into mesh at $t$ and take the difference between them. To compute the rollout error metrics, we always interpolate the predicted mesh to the ground-truth mesh at each step, and compute the metrics in the ground-truth mesh.

### B.4 BASELINES

Here we provide additional details on the baselines used in 2D mesh-based simulation. All the baselines use pre-trained evolution function $f_\theta^{\text{evo}}$ as part of the respective forward models. In the following, we mainly give details on functions responsible for choosing edges to split and coarsen which correspond to the policy function $f_\varphi^{\text{policy}}$ in LAMP's forward model.

**MeshGraphNets + heuristic remeshing**. At each step, instead of having $f_\varphi^{\text{policy}}$ to infer probability on edges to split and coarsen, we compute local curvature on edges of the mesh and use the curvature to filter out edges to split. Here, the local curvature on an edge is defined as the angle made by two unweighted normal vectors on boundary nodes of the edge (the normal vector on nodes is defined as the mean of normal vectors on faces surrounding the nodes). When we filter out edges in the mesh, we first choose edges with curvature exceeding pre-defined threshold, which is set to be 0.1 in the experiment. We next check that after splitting, the edges are not shorter than a pre-defined minimum length, which is set to be 0.04, to avoid having exceedingly small faces. Edges violating either of these criteria are not chosen as edges to split.

**MeshGraphNets + GT remeshing**. In this baseline, we use the mesh configuration of ground-truth meshes provided by the classical solver as the mesh configuration of predicted meshes. Specifically, after having $f_\theta^{\text{evo}}$ predict a mesh at the next time step, we interpolate the predicted mesh into the ground-truth mesh. We generate all the ground-truth meshes with the same initial condition as LAMP's rollout experiment.

**LAMP (no remeshing)**. This baseline does not involve any functions responsible for remeshing and we just recursively apply $f_\theta^{\text{evo}}$ to evolve meshes. Therefore, mesh topology of all meshes in a trajectory is isomorphic.

## C ADDITIONAL RESULT VISUALIZATION

### C.1 1D NONLINEAR PDES

In this section, we provide additional randomly sampled results for the 1D nonlinear PDE experiment, as shown in Fig. 8. We see that similar to Fig. 2, our LAMP is able to add more vertices to the locations with more dynamicity, while remove more vertices at more static regions. Moreover, with increasing $\beta$ that emphasizes more on reducing computational cost, LAMP splits less and coarsens more.

### C.2 2D MESH-BASED SIMULATION

In this section, we provide additional randomly sampled results for the 2D mesh-based simulation, as shown in Fig. 9. We see that our LAMP (fifth column) learns to add more edges to locations with higher curvature, resulting in better rollout performance than with no remeshing.

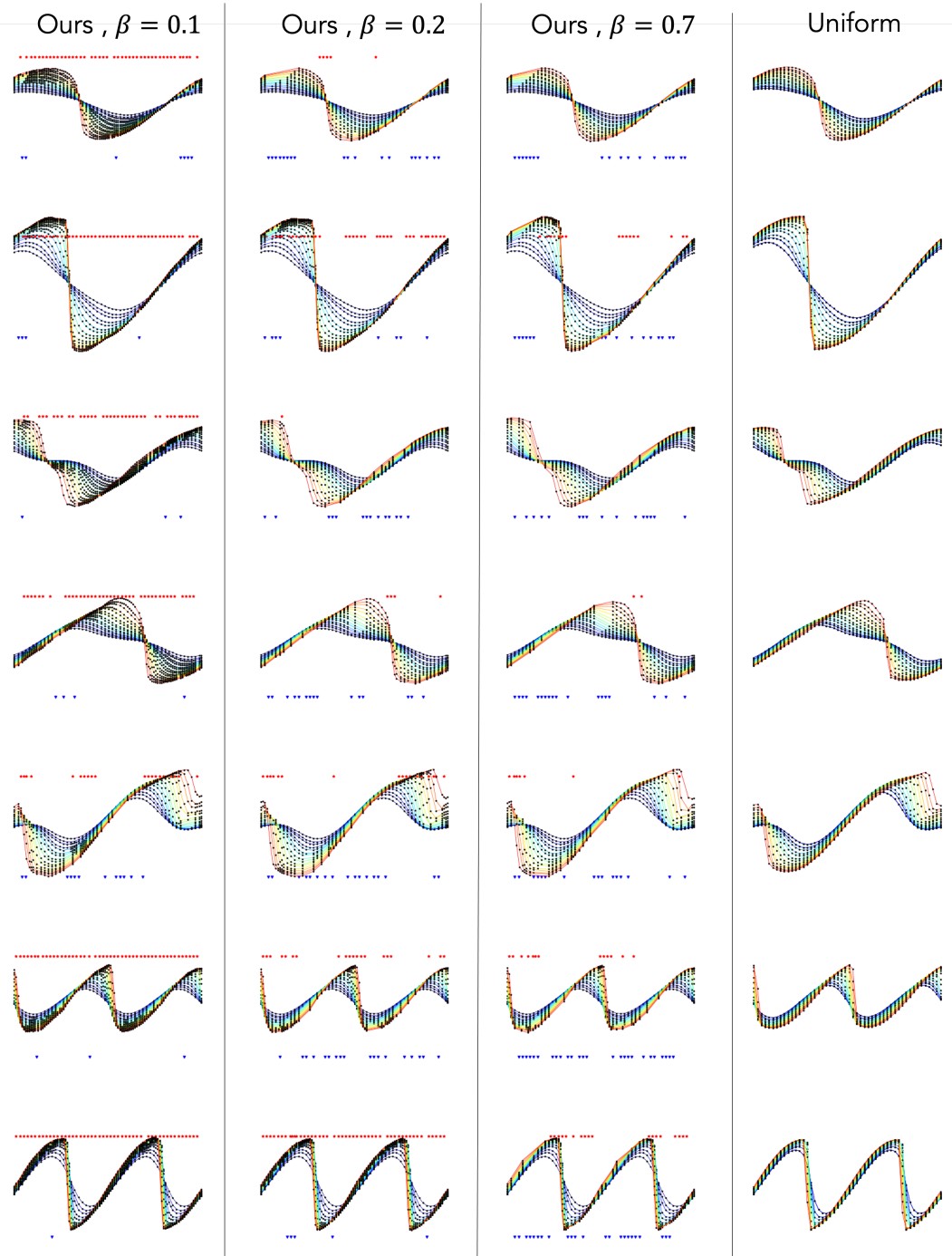

Figure 8: Additional example rollout results with different $\beta$ of our LAMP on 1D nonlinear PDEs with initial state on 50 vertices which is uniformly downsampled from the 100 vertices. The rollout is performed over 200 time steps, where different color denotes the system's state at different time. The upper red dots and lower blue dots shows the added and removed nodes of the mesh, comparing the end and the initial mesh.

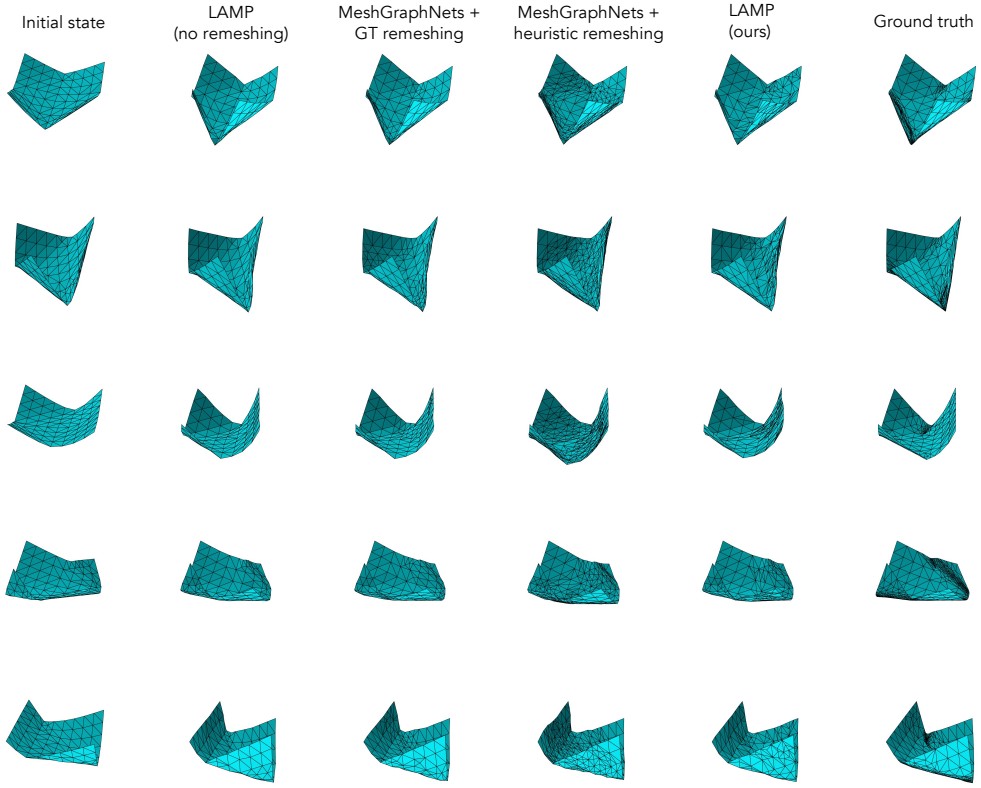

Figure 9: Additional example rollout results for 2D paper folding, at $t = 20$. We see that our LAMP (fifth column) learns to add more edges to locations with higher curvature, and learns to coarsen on the flat region (third and fourth row of "LAMP (ours)", where we see coarsening in the middle of our meshes), resulting in better rollout performance than with no remeshing and other baselines.

