# OpenReview forum: "Learning Controllable Adaptive Simulation for Multi-resolution Physics"
_ICLR.cc/2023/Conference — ICLR 2023 notable top 25%_

### Official Review · Reviewer_tR71 · 2022-10-15

**Confidence:** 4
**Correctness:** 3
**Technical Novelty And Significance:** 2
**Empirical Novelty And Significance:** 2
**Recommendation:** 6

**Clarity, Quality, Novelty And Reproducibility:**

In general the paper is quite easy to follow. On the other hand, it has a few important limitations:

A. Literature.

The paper did not reference and/or compare with recent paper "Reinforcement Learning for Adaptive Mesh Refinement" (https://arxiv.org/abs/2103.01342), which focus on the same applications (DRL for AMR). I would strongly encourage the authors to make a formal comparison with the work.

B. To further verify the approach, more thorough experiments need to be performed.

1. From the paper, it is clear that re-meshing is important but I am not sure whether RL is necessary. Did the author compare LAMP with simple heuristics (e.g., if the velocity in a local neighborhood is high, then it is obvious that the neighborhood requires some refinement)?  How does LAMP deal with PDE with large stiffness, e.g., the dynamics of a piece of cloth contacting with an immovable support. Will LAMP discover the stiffness in time and refine the mesh, and does it do better than existing re-meshing heuristics used in adaptive solvers (e.g., ZZ policy mentioned in https://arxiv.org/abs/2103.01342)? Given the current evaluation, there is no convincing evidence that RL is necessary.

2. What's the gap between LAMP and the ground-truth simulation using very fine-grained mesh? Table 1 shows that the number of vertices is up to 100. How does this approach scale to simulation of larger scale?

C. Some important details are missing, which could prevent audience unfamiliar with the literature from fully understanding this paper. For example:

1. The definition of action space is quite vague. While I see the definition of refinement/coarsening action in the appendix for 2D simulation, many details still remain unclear. Are split+flip together counted as "refinement"? Why "two edges on the same face of the mesh cannot be refined at the same time, nor can they be both coarsened"? What's the intuition? Can an example be given? In the equation of Sec 3.1 (under action representation), how the conditional probability $p(K^{re})$ and $p(K^{co})$ are evaluated, are you using softmax? How does summation over random variables $p(K^{re})$ and $p(K^{co})$ work? It is not clear at all from the text. Would be good to put a figure to explain. Also, how these actions defined in 1D still remain unclear.

2. In Sec. 3.2, the paper mentioned that the ground truth is computed with ground-truth solver "with very fine-grained mesh". Does the ground-truth simulation start from initial time step t = 0? What if the mesh predicted by the current policy is very far from the ground truth? If the simulation starts from t (and execute for S steps), then will such on-policy ground truth evaluation lead to very slow training process? Unfortunately, I didn't see any evaluation regarding to the time cost.

3. It is not clear how the training and evaluation PDE distribution are picked. Is the performance reported as in-distribution or out-of-distribution?

Due to clarity issues in the paper, it may not be able to be reproduced easily.

**Strength And Weaknesses:**

Strength
1. The paper is well-written on the high-level and easy to follow.
2. The idea is straightforward and clearly presented.

Weakness
1. The paper missed important literatures to compare against.
2. To further verify the approach, more thorough experiments need to be performed.
3. Some important details are missing, which could prevent audience unfamiliar with the literature from fully understanding this paper.



**Summary Of The Paper:**

The paper learns a policy to automatically re-mesh the current mesh used in physics simulation, so that delicate regions where rich dynamics happens can leverage more computational power. It uses reinforcement learning to train the policy end-to-end, with the reward being a combination of simulation accuracy and speed. The resulting method, called LAMP, is evaluated on simple 1D PDE and 2D mesh-based paper simulation.

**Summary Of The Review:**

Overall due to these important limitations, I cannot recommend acceptance.

======

UPDATE: the authors have made substantial efforts and addressed most of my concerns. I really appreciate. Thanks!

As a result, I raised the score to 6 and will not object acceptance of this paper in ICLR.

---

> ### Author Response · Authors · 2022-11-18
> **Official Response (4)**
>
> > Re8: In Sec. 3.2, the paper mentioned that the ground truth is computed with ground-truth solver "with very fine-grained mesh". Does the ground-truth simulation start from initial time step t = 0? What if the mesh predicted by the current policy is very far from the ground truth? If the simulation starts from t (and execute for S steps), then will such on-policy ground truth evaluation lead to very slow training process? Unfortunately, I didn't see any evaluation regarding to the time cost.
>
> Answer: Thanks for the question. Yes, all the ground-truth trajectories are generated by the ground-truth solver starting from time step t=0. To clarify, we don’t need to use a ground-truth simulator during the process of training. We first pre-generate $N$ ground-truth training trajectories on the fine mesh with a classical solver and save it into the file. These pre-generated trajectories will be loaded in the training, so that no ground-truth solver is needed during training LAMP. During the training, we sample an initial state from those pre-generated trajectories and perform remeshing and forward prediction. The predicted state is then interpolated to the ground-truth mesh using $g^{\text{interp}}(⋅)$ (see Appendix B.2 and B.3 of the updated manuscript) and then compared to the ground-truth trajectory. Therefore, no matter which time step the simulation starts from, the on-policy ground-truth evaluation takes the same amount of time - which is the time of $g^{\text{interp}}(⋅)$. Regarding the time cost for 2D, 1.2 million steps take around 1.5 days on an NVIDIA A100 80GB GPU. We hope this addresses the reviewer’s question.
>
> > Re9: It is not clear how the training and evaluation PDE distribution are picked. Is the performance reported as in-distribution or out-of-distribution?
>
> Answer: Actually, in the original submission, we already had the details of 1D dataset in section 4.1 of the main text and also 2D dataset in section 4.2 and the appendix A. All the results reported in the paper are in-distribution as we generate both training and test datasets with the same function class (but with different initial state and forces, drawn from the same distribution). For the 1D experiment, we first generate a dataset by changing static parameters and downsample from the original dataset to a smaller number of nodes. In a 2D mesh-folding experiment, we generate the dataset with a ground-truth solver by changing the kinematics added to the corners of papers. For the additional details, please also see sections 4.1 and 4.2 of the main text and the paragraph starting with "Generating data" of appendix B.3 in our updated manuscript.
>
> > Re10: Due to clarity issues in the paper, it may not be able to be reproduced easily.
>
> Answer: thanks for the above questions, in the updated manuscript, we have significantly improved Appendix A and B to give a comprehensive description about the architecture and training, incorporating answers to the questions you raised above. We will also release the code and data upon publication of the paper,  as stated in the updated Sec. 6 Reproducibility Statement. We hope that these address your concern about reproducibility.
>
> **Summary:**
>
> Through the above responses, we hope that we have addressed the reviewer's concerns and answer the reviewer's questions. Please let us know if there are further questions.

---

> ### Author Response · Authors · 2022-11-18
> **Official Response (3)**
>
> > Re6: The definition of action space is quite vague. While I see the definition of refinement/coarsening action in the appendix for 2D simulation, many details still remain unclear. Are split+flip together counted as "refinement"? Why "two edges on the same face of the mesh cannot be refined at the same time, nor can they be both coarsened"? What's the intuition? Can an example be given?
>
> Answer:
> Thanks for raising this question. We have updated the manuscript to make the description of the action space more clear. We answer the reviewer’s specific questions as follows. The action space for RL is defined as the product of two different sets: splitting and coarsening. Elements of each set indicate edges in a mesh and the policy function chooses some of the elements as edges to split or coarsen.  What we mean by “refinement” is performing split action on edges chosen by the policy function and remeshing means to perform split/flip/coarsen actions.
>
> The reason why “edges on the same face of the mesh cannot be refined at the same time” is that we may have an invalid mesh with some non-triangular faces such as quadrilaterals. Such an example is obtained as follows: suppose that we have a triangular face ABC and split edges AB and AC (see Fig. 6 in Appendix B.3 in the updated manuscript). When we denote the midpoints of AB and AC by D and E,  we have new edges DC and EB. If F denotes the intersection of DC and EB, we will have a quadrilateral ADFE. In order to avoid this situation, remeshing/coarsening action is basically performed on up to one edge of every face. This example is added in the Appendix B.3, under “Invalid remeshing”, in the updated manuscript.
>
> > Re7: In the equation of Sec 3.1 (under action representation), how the conditional probability p(K^re)  and p(K^co) are evaluated, are you using softmax? How does summation over random variables p(K^re) and p(K^co) work?
>
> Answer: Thanks for asking this question. We have updated the “action representation” in the main text to make this clearer. To clarify the reviewer’s question, we have a hyperparameter $K^{\text{max}}$ that defines the maximum number of edges to split or coarsen at each time step. For splitting, the policy network has a head whose output is the probability for the categorical distribution of {$0,1,2,...K^{\text{max}}$}, modeled via a **softmax** (answering the reviewer’s first question). The policy first samples a specific $K^{\text{re}}$ according to this categorical distribution, and obtains the log-probability of $K^{\text{re}}$, which is $\text{log} (K^{\text{re}}|M^t)$. Then, based on the sampled $K^{\text{re}}$, the policy will sample $K^{\text{re}}$ edges $(e_1^{\text{re}}, e_2^{\text{re}}, ..., e_{K^{\text{re}}}^{\text{re}})$ to split (each $e_k^{\text{re}}$ is the ID of an edge), and obtain the log-probability of choosing the specific $K^{\text{re}}$ edges. The joint log-probability for this specific splitting action is then given by:
>
> $\text{log}(K^{\text{re}}|M^t) + \sum_{k=1}^{K^{\text{re}}} \text{log}\ p(e_k^{\text{re}}|M^t).$
>
> Note that this log-probability is for the **specific sampled action** $(K^{\text{re}}, e_1^{\text{re}}, e_2^{\text{re}},..., e_{K^{\text{re}}}^{\text{re}})$. To answer the reviewer’s second question, there is **no summation** over the probability $\text{log}(K^{\text{re}}|M^t)$, and the summation is only over the log-probability of $K^{\text{re}}$ sampled edges to split. Similar reasoning is also applied to coarsening. Thus, we arrive at the expression under the “action representation”.

---

> ### Author Response · Authors · 2022-11-18
> **Official Response (2)**
>
> > Re3: How does LAMP deal with PDE with large stiffness, e.g., the dynamics of a piece of cloth contacting with an immovable support. Will LAMP discover the stiffness in time and refine the mesh, and does it do better than existing re-meshing heuristics used in adaptive solvers (e.g., ZZ policy mentioned in https://arxiv.org/abs/2103.01342)?
>
> Answer: To our understanding, the 2D mesh-based simulation does not meet some requirements, e.g., being second-order elliptic PDEs, for the superconvergence patch recovery technique which is involved in the derivation of ZZ estimator (proposed in [2]). Therefore, it’s unfeasible to compare LAMP with the LAMP+ZZ heuristics on the experiments. On the other hand, in the updated manuscript, we have added two baselines: MeshGraphNets + ground-truth remeshing (remeshing provided by the Adaptive Anisotropic Remeshing that is suitable for this simulation) and MeshGraphNets + heuristic remeshing, and showed that our LAMP outperforms both strong baselines. This demonstrates the effectiveness of LAMP (see also responses Re1 and Re2 above). For the stiffness, we recognize the importance to show LAMP’s scalability to other problems including PDEs with large stiffness, which we leave as a future research topic. As a part of the node feature, we use a one-hot vector to let the forward model distinguish objects that will add force from others receiving the force, which is also adopted in MeshGraphNets applied to dynamics in which a piece of cloth contacts with an object with stiffness in order to distinguish those objects. We expect that our model is able to recognize objects with stiffness and avoid remeshing on the objects.
>
>
> [2] Zienkiewicz, Olgierd Cecil, and Jian Zhong Zhu. "The superconvergent patch recovery and a posteriori error estimates. Part 1: The recovery technique." International Journal for Numerical Methods in Engineering 33.7 (1992): 1331-1364.
>
>
> > Re4: What's the gap between LAMP and the ground-truth simulation using very fine-grained mesh?
>
> Answer: The gap (difference) between LAMP and the ground-truth simulation using very fine-grained mesh consists of two components. One component is due to the difference between node features defined on the mesh predicted by LAMP’s evolution model and ground-truth meshes. Before RL training, we pre-train the evolution model on fine-grained ground-truth meshes so that the model performs reasonable rollout. This prediction error is minimized while learning the policy function as well. The second component is the error between the mesh topology predicted by the LAMP’s policy function and ground-truth meshes. The above two components result in the gap (difference) between simulation predicted LAMP and the ground-truth simulation.
>
> > Re5: Table 1 shows that the number of vertices is up to 100. How does this approach scale to simulation of larger scale?
>
> Answer: Thanks for raising the point of scalability. Besides the 1D experiment with up to 100 nodes, in the 2D mesh-based simulation in the updated manuscript (Table 2, which is also shown in the Re2 above), the number of nodes is up to **235** nodes, and the average number of nodes is 123.1. Our LAMP outperforms baselines of “MeshGraphNets + GT remeshing”,  “MeshGraphNets + heuristic remeshing”, and ablation “LAMP (no remeshing)”. This shows that LAMP is able to scale to larger, more complex mesh-based simulations.

---

> ### Author Response · Authors · 2022-11-18
> **Official Response (1)**
>
> We thank the reviewer for the thoughtful comments. Below we address the reviewer’s concerns.
>
> > Re1: The paper did not reference and/or compare with recent paper "Reinforcement Learning for Adaptive Mesh Refinement" (https://arxiv.org/abs/2103.01342), which focus on the same applications (DRL for AMR). I would strongly encourage the authors to make a formal comparison with the work.
>
> Answer: Thank you for the suggestion of the literature. In the updated manuscript, we have added the citation to the paper "Reinforcement Learning for Adaptive Mesh Refinement" [1] in the “Related Work” section. While [1] uses RL to learn the remeshing, it has notable differences with our work which make it not feasible to compare with in our experiment. Firstly, the method in [1] is evaluated for the specific finite element method, and cannot be directly applied for more general simulations, e.g., cloth simulations as in our experiment. Secondly, the actions of [1] are refinement on the **faces** of the **rectangular** meshes, which cannot be applied to our setting where the actions are refinement and coarsening on the **edges** of **triangular** meshes. Thus, the method proposed in [1] cannot be applied in our experiment. Moreover, [1] and our work have different goals: the goal of [1] is to reduce error, ours is to learn a controllable tradeoff between reducing error and reducing the computational cost. In the updated manuscript, we have provided a detailed discussion of [1] in the “Related Work” section, and made it clear the differences between the two works.
>
> [1] Yang, Jiachen, et al. "Reinforcement learning for adaptive mesh refinement." arXiv preprint arXiv:2103.01342 (2021).
>
> > Re2: From the paper, it is clear that re-meshing is important but I am not sure whether RL is necessary. Did the author compare LAMP with simple heuristics (e.g., if the velocity in a local neighborhood is high, then it is obvious that the neighborhood requires some refinement)?
>
> Answer: Thank you for the suggestion. In the updated manuscript, we have added an additional experiment that compares LAMP with “MeshGraphNets + heuristic remeshing” baseline, which demonstrates the necessity of RL. Here, the heuristic is to refine the edge according to the local curvature defined by the angle of normal vectors on the edge’s boundary nodes (the normal vector on a node is defined as the mean of normal vectors on faces surrounding the nodes). The intuition behind the use of curvature is that higher curvature regions should have more fine-grained faces. Note that the heuristic of using velocity in the local neighborhood to refine is not appropriate, since during folding, the crease of the folding region may have a smaller velocity than the flat part that is away from the crease (since the velocity = angular velocity * distance to the crease). Thus, we adopt the heuristic that is based on the local curvature.
>
> As we can see in the updated Table 2 and also shown below, our LAMP outperforms the “MeshGraphNets + heuristics remeshing” baseline by a wide margin, demonstrating that the use of RL is necessary. We can also see from the Fig. 4 and Fig. 9 of the updated manuscript that LAMP puts less number of finer faces into high curvature regions than the heuristic remeshing while achieving higher accuracy. This figure suggests that LAMP is able to optimize the number of finer faces added to the mesh while achieving accurate prediction.
>
> Model | Initial # vertices | Average # vertices | Error (MSE)
> :--: | :--: | :--: | :--:
> MeshGraphNets + GT remeshing | 102.9 | 115.9 | 5.91e-4
> MeshGraphNets + heuristic remeshing | 102.9 | 191.9 | 6.38e-4
> LAMP (no remeshing) | 102.9 | 102.9 | 6.13e-4
> LAMP (ours) | 102.9 | 123.1 | **5.80e-4**

---

### Official Review · Reviewer_WPia · 2022-10-23

**Confidence:** 3
**Correctness:** 3
**Technical Novelty And Significance:** 3
**Empirical Novelty And Significance:** 2
**Recommendation:** 5

**Clarity, Quality, Novelty And Reproducibility:**

Clarity
This paper is a little be hard to follow for me. There are so many different symbols in Section 3. I’m wondering whether a high-level diagram could improve its presentation. Also, Figure 3 should be important, but I did not find where it is referred in the main paper.

Quality
The experiments lack an important comparison with MeshGraphNet, which is the backbone of this work, and it can also learns to remesh.

Moreover, only two experiment cases are shown here (1D PDE and 2D square cloth). The authors could also use examples demonstrated in MeshGraphNet to make their evaluation more solid and comprehensive.

Furthermore, will the scale of the experiments be too small? In Section 3.1, the authors use Nedge = 1000 as an example. In contrast, the experiments only have ~100 edges.

Novelty
The concept of adaptive and adjustable remeshing from learning looks exciting to me. The novelty seems okay to me, as long as the paper could really demonstrate the benefit and effectiveness of its proposed method.

Reproducibility
There are many components (networks, RL, remeshing modules, etc) and hyperparameters in this pipeline, which are described as abstract symbols. I am afraid it might be hard to reproduce unless the authors can release their code.


**Strength And Weaknesses:**

Strength
This method models the adaptive mesh simulation as a multi-objective optimize and provides users with a convenient way to modify the relative importance of computational cost and simulation fidelity.

Moreover, the way they handle remeshing and treat the computational cost as a objective function is interesting. It could probably be generalized to other relative tasks other than simulation, e.g. inverse rendering with adaptive remsheing.

Weakness
First, the keyword `multi-scale` in the title seems not very appropriate to me. Even though this method can adaptively change the resolution in some local areas, all the dynamics are governed by the same evolution network. There are no multiscale or hierarchical structures as I can see. Moreover, according to the experiment results, the number of vertices does not change a lot (33->37.6, 50->53.2, 100->100.0, and 81.0->97.9). It is rather refinements than multi-scale physics.

The biggest concern for me is the experiment part. MeshGraphNet is its backbone and it also should be an important baseline, since MeshGraphNet itself can perform remeshing. However, it is not compared against MeshGraphNet.

Moreover, the paper also mentioned that it wants to verify ‘Can LAMP improve the Pareto frontier of Error vs. Computation, compared to state-of-the-art deep learning surrogate models’. But I did not find this experiment. Figure 3 is not referred to in the main text. I did not understand what does this figure mean. If possible, I’d like to see whether the Pareto frontier of this method can cover the performance of MeshGraphNet.


**Summary Of The Paper:**

The paper proposes to jointly learn the dynamics as well as a remeshing scheme for simulation based on discretized (triangular mesh) space. To consider both the simulation accuracy and the computational cost, this problem is formulated as a multi-objective optimization. Two sets of graph neural networks are used for learning the forward evolution and the remeshing policy, respectively. Experiments show that the relative importance of simulation error vs. computational cost can be adjusted by choosing different weight values during the training.

**Summary Of The Review:**

In summary, I think this paper is tackling an important problem in an interesting way. However, the experiments are not sufficient to really validate the advantage of their method. I hope the users can add more test cases and compare their method with MeshGraphNet. The composition of this paper can also be improved.

---

> ### Author Response · Authors · 2022-11-18
> **Official Response (3)**
>
> > Re3: Moreover, the paper also mentioned that it wants to verify ‘Can LAMP improve the Pareto frontier of Error vs. Computation, compared to state-of-the-art deep learning surrogate models’. But I did not find this experiment. Figure 3 is not referred to in the main text. I did not understand what does this figure mean. If possible, I’d like to see whether the Pareto frontier of this method can cover the performance of MeshGraphNet.
>
> Answer: Actually, in the original submission, our experiments in Table 1 had shown that the LAMP improves the Pareto frontier. To illustrate this in a more clear way, in the updated manuscript, we have also added the baselines (CNN, FNO, MP-PDE, and MeshGraphNets) to Figure 3. In the updated Figure 3, the blue solid markers are our LAMP, and the hollow markers are baselines. We see that LAMP does improve the Pareto frontier, by achieving similar error with much **less number of vertices**.
>
> > Re4: Clarity This paper is a little be hard to follow for me. There are so many different symbols in Section 3. I’m wondering whether a high-level diagram could improve its presentation. Also, Figure 3 should be important, but I did not find where it is referred in the main paper.
>
> Answer: We thank the reviewer for the suggestions on the writing and the presentation. Figure 1 serves as a high-level diagram of our whole pipeline and we have referenced it in section 3 of the submission. In the updated manuscript, we have added additional references to Figure 1 in section 3 and ensured that the symbols are consistent between the figures and the texts. We have also added references to Figure 3 in the main text where we discuss the “Pareto frontier of Error vs. Computation”. We hope these updates will improve the overall presentation.
>
> > Re5: Furthermore, will the scale of the experiments be too small? In Section 3.1, the authors use Nedge = 1000 as an example. In contrast, the experiments only have ~100 edges.
>
> Answer: Thanks for raising the point of scalability. Besides the 1D experiment with up to 100 nodes, in the 2D mesh-based simulation in the updated paper, the number of nodes is up to **235** nodes, and the average number of nodes is 123.1. This shows that LAMP is able to scale to larger, more complex mesh-based simulations. We left the extension to a larger scale problem for future works.
>
> > Re6:Reproducibility There are many components (networks, RL, remeshing modules, etc) and hyperparameters in this pipeline, which are described as abstract symbols. I am afraid it might be hard to reproduce unless the authors can release their code.
>
> Answer: We thank the reviewer for the point on reproducibility. We have added a reproducibility statement (Sec. 6), and we will make the code and data publicly available upon publication of the paper. Thanks to the reviewers’ suggestions, we have also updated Appendix A and B which detail the architecture and experiment setup, significantly improving the reproducibility of the paper.

---

> ### Author Response · Authors · 2022-11-18
> **Official Response (2)**
>
> > Re2: The biggest concern for me is the experiment part. MeshGraphNet is its backbone and it also should be an important baseline, since MeshGraphNet itself can perform remeshing. However, it is not compared against MeshGraphNet.Quality The experiments lack an important comparison with MeshGraphNet, which is the backbone of this work, and it can also learns to remesh.
>
> Answer: Thanks for raising the point of baseline comparison with MeshGraphNets + remeshing. In the updated manuscript, we have updated Table 2 by adding two more baselines. One is the baseline of “MeshGraphNets + GT remeshing”, where the “GT (ground-truth) remeshing” means that the remeshing is provided by the classical solver with Adaptive Mesh Refinement. Here we use ground-truth remeshing since it provides a lower rollout error than learned remeshing (since learned remeshing in MeshGraphNets needs to also predict sizing field which can result in additional error), as is shown in the MeshGraphNets paper. Another baseline we added is “MeshGraphNets + heuristics remeshing” according to reviewer tR71, where the heuristics is to refine the edge according to the local curvature. To address the point of scalability in the next question, in the updated Table 2, we now use initial states at t=10,30,50 (instead of previous t=0 only) and each roll out for 20 steps (compared to previous 10 steps). This has a larger initial and average mesh size, and is able to better test the scalability and longer-term prediction of our method.
>
> Below is updated Table 2 for 2D mesh-based simulation with added baselines:
>
> Model | Initial # vertices | Average # vertices | Error (MSE)
> :--: | :--: | :--: | :--:
> MeshGraphNets + GT remeshing | 102.9 | 115.9 | 5.91e-4
> MeshGraphNets + heuristic remeshing | 102.9 | 191.9 | 6.38e-4
> LAMP (no remeshing) | 102.9 | 102.9 | 6.13e-4
> LAMP (ours) | 102.9 | 123.1 |  **5.80e-4**
>
> From the updated table, we see that our LAMP outperforms both baselines and the no-remeshing ablation. Specifically, LAMP outperforms the strong baseline of “MeshGraphNets + GT remeshing”, where the remeshing is provided by the ground-truth Adaptive Mesh Refinement. This shows that LAMP can further improve upon MeshGraphNets + remeshing to learn a better remeshing policy, allowing the evolution model to evolve the system in a more faithful way. Furthermore, the “MeshGraphNets + heuristic remeshing” baseline has a larger error, showing that this intuitive baseline is suboptimal. Finally, LAMP outperforms its ablation without remeshing, showing the necessity of remeshing which can significantly reduce prediction error.

---

> ### Author Response · Authors · 2022-11-18
> **Official Response (1)**
>
> We thank reviewer WPia for the feedback and appreciate that reviewer finds our approach interesting and the problem to solve is significant. Below, we will address the reviewer’s concern on baseline comparisons, the Pareto frontier, and scalability.
> > Re1: First, the keyword multi-scale in the title seems not very appropriate to me. Even though this method can adaptively change the resolution in some local areas, all the dynamics are governed by the same evolution network. There are no multiscale or hierarchical structures as I can see. Moreover, according to the experiment results, the number of vertices does not change a lot (33 &rarr; 37.6, 50 &rarr; 53.2, 100 &rarr; 100.0, and 81.0 &rarr; 97.9). It is rather refinements than multi-scale physics.
>
> Answer: We thank reviewer WPia for the question. First, by “multi-scale”, we refer to system’s **multi-scale dynamics**, not **multi-scale modeling**. Specifically, what we mean by multi-scale dynamics is an **intrinsic nature** of the system that has vastly different dynamicity at different spatial locations (as explained in the introduction). Such heterogeneous dynamical systems are common in simulation and appear in many real-world problems, e.g., weather forecasting, plasma dynamics, materials sciences, and require different spatial resolutions at different locations to resolve the whole system accurately and efficiently. On the other hand, “multi-scale modeling” typically means that the system is **modeled** as two or more hierarchies. In our manuscript and title, what we mean by “multi-scale” is the former, i.e. multi-scale dynamics.
>
> Secondly, in our experiments, we choose two typical systems in 1D and 2D mesh with multi-scale dynamics. The number (33 &rarr; 37.6, 50 &rarr; 53.2, 100 &rarr; 100.0, and 81.0 &rarr; 97.9) are the average number of vertices over unrolled timesteps, and the mesh is refined and coarsened at each time step. As shown in Figure 2’s second subfigure (β=0.2), there are in total 21 edges refined and 8 edges coarsened, so the final mesh has only increased 21-8=13 edges/nodes compared to the initial mesh. Furthermore, there are 128 test trajectories, and some have net increase and some have net decrease of edges (for more example visualizations, see Figure 8 in the Appendix in the updated manuscript). Therefore, even though we don’t see a big change in the average number of vertices, each individual sample may have many refinements and coarsening. We can also see such multi-scale dynamics in Figure 4 and Figure 9 for 2D mesh simulation. Based on the above, we believe that the keyword “multi-scale” in the title is appropriate.

---

> ### Comment · Reviewer_WPia · 2022-11-29
> **Thank authors for the detailed response**
>
> I really appreciate the authors' detailed response but still have some concerns after reading the rebuttal.
>
> 1. The definition of multi-scale physics is not convincing.
>
> Even the classical solver (Adaptive Anisotropic Remeshing for Cloth Simulation), which is used as Ground Truth in the comparison, did not claim to solve multi-scale physics. Instead, I think their use of "multiresolution" is more appropriate. Can the authors provide more references about their definition?
> > First, by “multi-scale”, we refer to system’s multi-scale dynamics, not multi-scale modeling. Specifically, what we mean by multi-scale dynamics is an intrinsic nature of the system that has vastly different dynamicity at different spatial locations (as explained in the introduction). Such heterogeneous dynamical systems are common in simulation and appear in many real-world problems, e.g., weather forecasting, plasma dynamics, materials sciences, and require different spatial resolutions at different locations to resolve the whole system accurately and efficiently. On the other hand, “multi-scale modeling” typically means that the system is modeled as two or more hierarchies. In our manuscript and title, what we mean by “multi-scale” is the former, i.e. multi-scale dynamics.
>
> 2. The scale of the experiments.
>
> The remeshing operation is usually more important in fine-resolution cases with a large number of vertices. In cases where there are only < 250 vertices, the computation resources will not be a problem. Why not simply uniformly increase the resolution to improve the accuracy?
>
> Moreover, it is still misleading to use Nedge = 1000 as an example while the experiments are not at the scale. In small scales, 2^100 vs. 100 ^ 10 is not a very huge difference. In larger scales, that small number of remeshing operations (~10) is not significant to the entire simulation.

---

> ### Author Response · Authors · 2022-11-30
> **Response to the additional questions**
>
> We thank the reviewer for engaging in the discussion. Here are our responses and we hope this would address the concerns.
>
> > Re1: The definition of multi-scale physics is not convincing. Even the classical solver (Adaptive Anisotropic Remeshing for Cloth Simulation), which is used as Ground Truth in the comparison, did not claim to solve multi-scale physics. Instead, I think their use of "multiresolution" is more appropriate. Can the authors provide more references about their definition?
>
> Answer: Thanks for the suggestion. We agree that "multiresolution" is a word that more accurately captures the systems we simulate, where a small fraction of the system is highly dynamic, and requires very fine-grained resolution to simulate accurately, while a majority of the system is changing slowly. Thus, we will change “multiscale” to “multiresolution” in the title and in the main text, in the final version of the paper. We thank the reviewer for suggesting this change.
>
> > Re2: The scale of the experiments. The remeshing operation is usually more important in fine-resolution cases with a large number of vertices. In cases where there are only < 250 vertices, the computation resources will not be a problem. Why not simply uniformly increase the resolution to improve the accuracy?
>
> Answer: The purpose of our experiments is to demonstrate the effectiveness of our approach on both 1D and 2D problems, compared to alternative methods/baselines. In our experiments we demonstrated that our learned remeshing policy leads to a better forward prediction accuracy compared to without remeshing, remeshing with AMR, and remeshing with heuristics, which showed the effectiveness of our model. Since our GNN policy learns to predict the logit of actions on an edge based on the states of a local graph consisting of neighboring edges within a few hops. This policy is **shared** among all edges, independent of the total number of edges. Therefore, we expect that LAMP is able to scale to even larger mesh sizes. We left the extension to a larger scale problem for future works.
>
>
> > Re3: Moreover, it is still misleading to use Nedge = 1000 as an example while the experiments are not at the scale. In small scales, 2^100 vs. 100 ^ 10 is not a very huge difference. In larger scales, that small number of remeshing operations (~10) is not significant to the entire simulation.
>
> Answer: We understand the reviewer's concern. The number “1000” here was intended to serve as a number for demonstration. To address this, we will change the number to “200” in the paper (since the meshes in our experiments have up to 235 nodes), and 2^200=10^60 vs. 200^10=10^23 is still a huge difference.
>
> The choice of remeshing operations, K^max, is a hyperparameter which can be set by the practitioner prior to the training of the policy. In our experiments, we used K^max=20. We expect that for a larger initial mesh, a larger K^max is more appropriate. Suppose that the initial number of nodes is ~5000, and setting K^max=50 would probably suffice, since after 50 steps of rollout, LAMP can refine or coarsen up to 50*50=2500 edges, which is on the same order as 5000.

---

### Official Review · Reviewer_mWzR · 2022-10-24

**Confidence:** 3
**Correctness:** 3
**Technical Novelty And Significance:** 3
**Empirical Novelty And Significance:** 3
**Recommendation:** 8

**Clarity, Quality, Novelty And Reproducibility:**

The paper quality and organization as well as the novelty of the proposed method is good. However, reproducibility of the method is hard as the source-code and data were inaccessible at the time of review.

**Strength And Weaknesses:**

## Strengths:

The paper proposes a novel (LAMP) model which jointly learns to predict the evolution of a process of interest (using graph-neural network based architecture) as well as learns to adaptively refine / coarsen various parts of the network topology (I.e., mesh) of the domain being forecast. The proposed loss functions combining  RL policy learning task and domain evolution learning task are novel and indicated to be effective from the presented results.


The results demonstrate the power of the LAMP model over other state-of-the-art models for forecasting PDE trajectories including the benefit of mesh-refinement which is one of the novel proposals made by the current work. Further, the mesh refinement (measured in Table 1, as Avg. # vertices) does not seem to significantly increase the number of nodes (I.e., the refinement policy seems to be applied systematically without leading to an explosion of the problem complexity).


The problem formulation, especially as a balance between the number of FLOPs (computational cost of refinement / coarsening resulting form the RL policy actions) and long-term prediction error is novel and effective.


The paper is (mostly) clearly written and well organized but for a clear description of the experimental setup.

## Weaknesses:

My major problem with the paper is the lack of a cohesive / complete discussion of the experimental setup. The proposed modeling pipeline (RL + forecasting using graph neural networks where each of the forecasting and RL based losses further have multiple tasks optimized therein) has a non-trivial organization and many hyperparamteres / steps that are not described in detail. E.g., Eq. 4 the $g^{\mathrm{interp}}(\cdot)$ function does not find detailed mention in the discussion.


The paper does not compare with MeshGraphNets [1], which is the base model the paper employs to design its framework. Without this, it is hard to contextualize the degree of improvement of the prediction performance of the proposed framework. Although the current paper makes other contributions (I.e., adaptive-mesh-refinement based on learned policy) it is still important to understand how the evolution performance in the LAMP model fares w.r.t MeshGraphNets which is a closely related model.


Further, without access to the source code, it is non-trivial to fully comprehend the effectiveness of the proposed pipeline with so many moving parts.


## Questions for Authors:

1. How are $\alpha_s^{\mathrm{policy}}$, $\alpha_s^{\mathbb{I}}$ parameters (Eq. 10) set / learned?


2. How sensitive is the proposed model to values of $\beta$, $\alpha_s^{\mathrm{policy}}$, $\alpha_s^{\mathbb{I}}$?


3. What type of interpolation function is used for $g^{\mathrm{interp}}(\cdot)$ ? How many types of interpolations were tried, and which was the most effective? Also, how does the computational cost of $g^{\mathrm{interp}}(\cdot)$ figure into the cost of the overall learning model?


4. What is the reasoning behind the selection of an alternating strategy to train the evolution and policy losses?


5. According to Eq. 9, the $L^{\mathrm{evo}}_S$ term is comprised of a fine-grained mesh based forecasting loss (second term) and a forecasting loss based on the mesh generated by the RL policy. What would be the effect of decaying the second term I.e., fine-grained mesh based forecasting over the course of model training as this might have significant effects on decrease in computation time of training for large differences between fine-grained / coarse-grained meshes?


## References
1. Pfaff T, Fortunato M, Sanchez-Gonzalez A, Battaglia PW. Learning mesh-based simulation with graph networks. arXiv preprint arXiv:2010.03409. 2020 Oct 7.

**Summary Of The Paper:**

Authors propose a novel methodology based on Graph Neural Networks (and reinforcement learning) to predict the evolution of a process of interest. The primary context in which the proposed `LAMP`  (Learning Controllable Adaptive simulation for Multi-scale Physics) model is applied is to forecast the evolution of systems governed by PDEs and on a real-world task of predicting the evolution of deformation of paper with forces acting on it.  Overall, the main novelty of the proposed model is that the model is able to not only forecast the evolution of states of the process of interest, but is also able to adaptively yield fine-grained / coarse-grained predictions as needed based on the inferences made by a learnable adaptive-mesh-refinement policy network (also part of the LAMP pipeline).  The results demonstrate that LAMP yields results superior to powerful state-of-the-art PDE forecasting models.

After the authors' responses especially about the ablation study, I am relatively more convinced about the effectiveness about the proposed method and that the proposed method is indeed able to address an important problem of developing ML based surrogates of scientific simulations while also being able to perform adaptive meshing on the domain.

**Summary Of The Review:**

The proposed LAMP model is novel as it is able to effectively develop a graph neural network based model for forecasting evolution of non-linear PDE based domains along with adaptive-mesh-refinement as proposed by a learned RL policy (jointly learned with the evolution learning component of the LAMP pipeline).

---

> ### Author Response · Authors · 2022-11-18
> **Official Response (2)**
>
> In the following, we answer the reviewer’s questions.
>
> > Re4: How are $\alpha_s^\text{policy}$, $\alpha_s^\mathbb{I}$ parameters (Eq. 10) set / learned?
>
> Answer: We set both the value of $\alpha_s^\text{policy}$ and $\alpha_s^\mathbb{I}$ as 1, since this is the simplest, most natural choice. We find that this setting already has a good performance, so have not tried other combinations.
>
>
> > Re5: How sensitive is the proposed model to values of $\beta$, $\alpha_s^\text{policy}$, $\alpha_s^\mathbb{I}$?
>
> Answer: As stated in “Learning to adapt to varying $\beta$” in Section 3.2  and also in the updated Table 3, during training of the actor-critic, $\beta$ is sampled from a range, which we use [0,0.5] in our 1D and 2D experiments. At inference, when varying $\beta$, we see that LAMP can adaptively adjust how much it refines or coarsens, as shown in Figure 3. For $\alpha_s^\text{policy}$, $\alpha_s^\mathbb{I}$, we have not tested other combinations of other than the simplest, most natural choice of all 1s, since it already results in a good performance.
>
> > Re6: What type of interpolation function is used for  ginterp(⋅)? How many types of interpolations were tried, and which was the most effective? Also, how does the computational cost of ginterp(⋅)  figure into the cost of the overall learning model?
>
> Answer: We used barycentric interpolation for $g^{\text{interp}}$, which we detail in the Appendix B.2 and B.3 of the updated manuscript. It constructs the node features of the newly-added node by **linearly** interpolate the node features of neighboring nodes, according to their relative positions. We have only tried barycentric interpolation, since this is the simplest and is what the MeshGraphNets paper used. Having a higher-order interpolation will require more neighbors of the mesh and will significantly increase the complexity of the interpolation.
>
> In the paper, we use the number of vertices as a surrogate metric for measuring the computational cost, which has taken into account the computational cost of $g^{\text{interp}}(⋅)$. As stated under Eq. (13), both the evolution and the remeshing networks are GNNs, and the computational cost (in terms of FLOPs) for both GNNs scales linearly with the number of vertices. Since remeshing and evolution are done per step, the total computational cost will scale linearly with the number of vertices. Reducing the number of vertices proportionally reduces the computational cost of computing the remeshing.
>
> > Re7: What is the reasoning behind the selection of an alternating strategy to train the evolution and policy losses?
>
> Answer: The reasoning behind using alternating training strategy for actor-critic and evolution is as follows. When learning the actor-critic, the “return” for the RL is the expected reward based on the **current** policy and evolution model. If at the same time the evolution model is also optimized together, then the reward function will always be changing, which will likely make learning the policy harder. Therefore, we have adopted the alternating training strategy, which is also widely used in many other applications, such as in GAN training. We have added this explanation to the updated Appendix B.1.
>
> > Re8: According to Eq. 9, the LSevo term is comprised of a fine-grained mesh based forecasting loss (second term) and a forecasting loss based on the mesh generated by the RL policy. What would be the effect of decaying the second term I.e., fine-grained mesh based forecasting over the course of model training as this might have significant effects on decrease in computation time of training for large differences between fine-grained / coarse-grained meshes?
>
> Answer: If we understand correctly, the “decay” by the reviewer means reducing the effect of the second term, which can be done in a few ways. (1) If “decay” means multiplying the second term with a smaller coefficient, it will not reduce the training time, since the second term still needs to be computed. Therefore, there is no benefit in simply multiplying the second term with a smaller number. (2) If “decay” means removing the second term, it will definitely reduce the training time, but the evolution model will not be trained with very fine-grained mesh. Thus, in testing time, the evolution model may not adapt to very fine-grained meshes, if it is unrolled for many steps and the mesh becomes very fine-grained. In our early test, we found that having the second term improves the accuracy than without, so in all later experiments we always have the second term with a coefficient of 1.

---

> ### Author Response · Authors · 2022-11-18
> **Official Response (1)**
>
> We thank the reviewer for the constructive review, and are glad that the reviewer recognizes the novelty, significance, and effectiveness of our method. In the following, we address the points the reviewer raised about experimental setup and baseline comparison, and answer the questions.
>
> > Re1: My major problem with the paper is the lack of a cohesive / complete discussion of the experimental setup. The proposed modeling pipeline (RL + forecasting using graph neural networks where each of the forecasting and RL based losses further have multiple tasks optimized therein) has a non-trivial organization and many hyperparamteres / steps that are not described in detail. E.g., Eq. 4 the ginterp(⋅) function does not find detailed mention in the discussion.
>
> Answer:  Thanks for the suggestion. We have updated the manuscript in which we significantly expanded Appendix B, providing concrete details about the complete experiment setup. This includes training stages (pre-training of evolution model and joint training of actor-critic and evolution model), details in each stage, action space, interpolation function $g^{\text{interp}}$, and a complete hyperparameter table. Please refer to the updated manuscript for details. Moreover, as indicated in the response to Re3, we will also release the codebase and dataset upon publication.
>
> > Re2: The paper does not compare with MeshGraphNets [1], which is the base model the paper employs to design its framework. Without this, it is hard to contextualize the degree of improvement of the prediction performance of the proposed framework. Although the current paper makes other contributions (I.e., adaptive-mesh-refinement based on learned policy) it is still important to understand how the evolution performance in the LAMP model fares w.r.t MeshGraphNets which is a closely related model.
>
> Answer: Thanks for raising the point of baseline comparison with MeshGraphNets + remeshing. In the updated manuscript, we have updated Table 2 by adding two more baselines. One is the baseline of “MeshGraphNets + GT remeshing”, where the “GT (ground-truth) remeshing” means that the remeshing is provided by the classical solver with Adaptive Mesh Refinement. Here we use ground-truth remeshing since it provides a lower rollout error than learned remeshing (since learned remeshing in MeshGraphNets needs to also predict sizing field which can result in additional error), as is shown in the MeshGraphNets paper. Another baseline we added is “MeshGraphNets + heuristic remeshing” according to reviewer tR71, where the heuristic is to refine the edge according to the local curvature. To address the point of scalability in the next question, in the updated Table 2, we now use initial states at t=10,30,50 (instead of previous t=0 only) and each roll out for 20 steps (compared to the previous 10 steps). This has a larger initial and average mesh size, and is able to better test the scalability and longer-term prediction of our method.
>
> Below is updated Table 2 for 2D mesh-based simulation with added baselines:
>
> Model | Initial # vertices | Average # vertices | Error (MSE)
> :--: | :--: | :--: | :--:
> MeshGraphNets + GT remeshing | 102.9 | 115.9 | 5.91e-4
> MeshGraphNets + heuristic remeshing | 102.9 | 191.9 | 6.38e-4
> LAMP (no remeshing) | 102.9 | 102.9 | 6.13e-4
> LAMP (ours) | 102.9 | 123.1 |  **5.80e-4**
>
> From the updated table, we see that our LAMP outperforms both baselines and the no-remeshing ablation. Specifically, LAMP outperforms the strong baseline of “MeshGraphNets + GT remeshing”, where the remeshing is provided by the ground-truth Adaptive Mesh Refinement. This shows that LAMP can further improve upon MeshGraphNets + remeshing to learn a better remeshing policy, allowing the evolution model to evolve the system in a more faithful way. Furthermore, the “MeshGraphNets + heuristic remeshing” baseline has a larger error, showing that this intuitive baseline is suboptimal. Finally, LAMP outperforms its ablation without remeshing, showing the necessity of remeshing which can significantly reduce prediction error.
>
> > Re3: Further, without access to the source code, it is non-trivial to fully comprehend the effectiveness of the proposed pipeline with so many moving parts.
>
> Answer: We have added a reproducibility statement (Sec. 6), and we will make the code and data publicly available upon publication of the paper. Thanks to the reviewers’ suggestions, we have also updated Appendix A and B which detail the architecture and experiment setup, significantly improving the reproducibility of the paper.

---

### Official Review · Reviewer_vMEa · 2022-10-26

**Confidence:** 4
**Correctness:** 4
**Technical Novelty And Significance:** 4
**Empirical Novelty And Significance:** 3
**Recommendation:** 8

**Clarity, Quality, Novelty And Reproducibility:**

As far as the method, the paper is easy to follow and the approach is well motivated. The approach is clearly described, but might have missed some of the details on training the model. Specifically: How the episodes for the RL training are set up? Is f_evo pre-trained or trained jointly with the re-meshing? How to determine the number of action steps K? What K did you use?  During the training, do you compute the reward on the rollouts or after every time step *t*?

**Strength And Weaknesses:**


**Strengths**

The main advantage of the proposed approach is being to trained the re-meshing procedure in an unsupervised way, while the previous approach requires explicit supervision on the sizing field provided by the simulator or estimated (assuming that the GT simulations include re-meshing). The unsupervised approach can be particularly advantageous on the real data, where the information about ground-truth re-meshing is not available.

**Weaknesses**

1. Choice of K^max and evaluation on larger experiments.  If my understanding is correct, the model performs up to K^max actions of refinement and coarsening at each step of the simulation. Intuitively, K^max should be higher on the more fine mesh, as we might need to modify more edges. How was K^max chosen?

    The current paper evaluated the simulations only up to 100 nodes, while the previous MeshGraphNet paper demonstrated the results on up to ~5000 nodes. Do the authors think that the approach can scale to this number of nodes. How do you expect the choice of K^max to change, if we increase the number of nodes to 5000? Given that the action space is N_edge^K, do the authors think that it would be feasible to train the method with such a large action space?


2. The choice of baselines and comparison to the MeshGraphNet+remeshing
I am surprised with the choice of the baselines. The cloth mesh can be naturally represented as a graph, but the paper provides only one
one graph-net baseline (LAMP with not re-meshing). Using the CNN as a baseline enforces that the initial mesh should be represented as a grid, which is quite limiting.

    Also, the paper does not provide the comparison to the previous MeshGraphNet method *with re-meshing*, as it is the only other method that uses re-meshing. It is not an entirely fair comparison, as MeshGraphNets use supervision on re-meshing, but it would be useful to compare which parts of the mesh are modified, and how many edges are added/collapsed between the two approaches.

**A few questions**

1. It seems that the re-meshing process only adds 3-4 vertices on average (Table 1). I am surprised that it results in such a big change in MSE for 30 and 50 initial edges. Can the authors explain why this might be the case? Also, do you have a sense of how many edges were actually changed through re-meshing? For example, it is possible that some edges were added/collapsed, but the average number of vertices stayed the same.

2. Figure 4: LAMP seems to add new edges to a different regions than the ground-truth. Specifically, GT adds more edges on the cloth bending with the highest curvature, while LAMP adds the edges closer to the centre of the cloth. Can the authors provide the intuition why this might be the case and, perhaps, how to remedy this?

3. The RL-based approaches tend to require a lot of training episodes, especially if several edges are modified at each time step. How many episodes are roughly needed to train the policy and the value function. Also, is the model trained together with f_evo? Do you have a sense how the required number of episodes changes with the choice of K^max?


**Summary Of The Paper:**

This paper proposes uses an actor-critic for adapting the simulation mesh, specifically to add or collapse edges. The approach explicitly makes a trade-off between error and computational cost, allowing to tune it at test time.

**Summary Of The Review:**

Overall, learning how to modify the mesh in an unsupervised way using RL makes a lot of sense. It allows to 1) increase the simulation accuracy by adding vertices to the part of the cloth with high curvature 2) save computation time by removing vertices from the mesh with low curvature 3) unlike the previous MeshGraphNet approach, LAMP does not require to have the re-meshing sizing field from the ground-truth simulator, which is hard to obtain.

However, it will be helpful to have more insight in the choice of K^max and how the required number of coarsening/refinement steps is needed for different mesh sizes. It is also concerning that the approach adds only 3-4 vertices on average and does not add new vertices to the mesh with 100 nodes. Also, Figure 4 shows that the approach does not add the edges to the parts of the cloth with the highest curvature. It would be great to get author’s insights on why the might be the case. ~I am giving borderline accept, but willing to increase to accept if the authors demonstrate the ability to scale to meshes with >100 nodes and clarify why so few vertices need  to be added to the mesh.~

EDIT: the authors addressed all my concerns about scalability, provided more examples and explanation how the edges are refined/coarsened and added a relevant baseline with the MeshGraphNet. The updated version of the paper is convincing and provides value to the learning simulation community. I have increased the score to 8 (accept).

---

> ### Author Response · Authors · 2022-11-18
> **Official Response (6)**
>
> > Re6: It is also concerning that the approach adds only 3-4 vertices on average and does not add new vertices to the mesh with 100 nodes. Also, Figure 4 shows that the approach does not add the edges to the parts of the cloth with the highest curvature. It would be great to get author’s insights on why the might be the case. I am giving borderline accept, but willing to increase to accept if the authors demonstrate the ability to scale to meshes with >100 nodes and clarify why so few vertices need to be added to the mesh.
>
> Answer: Thanks for the summary of your concerns. In the above responses Re1 to Re5, we have hopefully addressed all of your concerns, which helps improve the paper. Importantly, in the 2D experiments in the updated manuscript, we demonstrate that LAMP is able to scale to >100 nodes (see above Re2), and we also explain why the appearance of few vertices are needed to be added to the mesh (see above Re3). Please let us know if you have further questions, and happy to provide answers.

---

> > ### Comment · Reviewer_vMEa · 2022-11-18
> > **Strong response from the authors.**
> >
> > I appreciate that the authors provided an extensive response to my questions and making the new experiments with MeshGraphNets and simulations with more nodes. With the updated version, I find the message of the paper very compelling, and I increased the score to 8.

---

> ### Author Response · Authors · 2022-11-18
> **Official Response (5)**
>
> > Re5: The RL-based approaches tend to require a lot of training episodes, especially if several edges are modified at each time step. How many episodes are roughly needed to train the policy and the value function. Also, is the model trained together with f_evo? Do you have a sense how the required number of episodes changes with the choice of K^max?
>
> > As far as the method, the paper is easy to follow and the approach is well motivated. The approach is clearly described, but might have missed some of the details on training the model. Specifically: How the episodes for the RL training are set up? Is f_evo pre-trained or trained jointly with the re-meshing? How to determine the number of action steps K? What K did you use? During the training, do you compute the reward on the rollouts or after every time step t?
>
> Answer: Thanks for raising these questions. In the updated manuscript, we have significantly expanded Appendix B, in which we detail the training procedure and provide a more detailed hyperparameter table (Table 3). In the following, we answer the reviewer’s specific questions.
>
> (1) “Is f_evo pre-trained or trained jointly with the re-meshing?”
>
> For the training procedure, there are two stages of training. The first stage is pre-training the evolution model alone without remeshing, and the second stage is alternative training of actor-critic and evolution model, i.e., alternatively switching between two phases: learning the policy with RL (while keeping the evolution frozen) and finetuning the evolution model (with the policy frozen).
>
> (2) “How the episodes for the RL training are set up?”
>
> For the second stage of alternative training, we use 30 epochs, where in each epoch there are 2016 episodes for 1D and 640 episodes for 2D experiment. Each episode is a minibatch of B trajectory chunks where each trajectory chunks is a consecutive $S$ number of steps: {$t, t+1, t+2, …t+S$} that we randomly sample from all the trajectories (for 1D, batch-size is 128, for 2D, batch-size is 64). We set up the number of epochs so that it is more than enough for learning the policy. See the answer to the question (3) for the number of episodes needed.
>
> (3) “How many episodes are roughly needed to train the policy and the value function?”
>
> We observe that after around 2000 episodes, the policy and value model of LAMP is learned reasonably, where the reward becomes positive and the LAMP outperforms the no-remeshing ablation in the hold-out validation set. With more episodes, the performance continues to improve.
>
> (4) “Do you have a sense how the required number of episodes changes with the choice of K^max”
>
> Previously we have tested with $K^{\text{max}}$ of 10 and 20 (current default), and don’t see a noticeable increase of required episodes to train, and we expect that with a larger $K^{\text{max}}$, the required number of steps will likely increase, but not by a large amount.
>
> (5) “How to determine the number of action steps K?”
>
> To clarify, $K^{\text{max}}$ is a hyperparameter, which sets up the maximum number of refinement or coarsening in one action. As detailed in the “action representation” in Section 3.1, the policy network first samples a $K$ from {$0,1,2,...K^{\text{max}}$}, and then performs $K$-steps of refinement or coarsening based on the log-probability of each edge. Note that the decision of $K$ from {$0,1,2,...K^{\text{max}}$} is also part of the action, where during training, the log-probability of this decision of choosing $K$ from {$0,1,2,...K^{\text{max}}$} is also optimized with RL. On the other hand, the choice of hyperparameter $K^{\text{max}}$ mainly depends on the practitioner’s need, so that within the rollout period, the maximum number of refinement or coarsening is on the same order as the number of nodes. For example, the average number of initial nodes for the 2D mesh is 102.9. We set $K^{\text{max}}$=20, so that within 20 steps of rollout, it is possible to refine or coarsen 20*20=400 number of edges, which is on the same order as 102.9. In our case, for both 1D and 2D, we use $K^{\text{max}}$=20, based on this principle.
>
> (6) During the training, do you compute the reward on the rollouts or after every time step t?
>
> During the training, we compute the reward after every time step $t$. Specifically, when training the actor-critic, for one episode with time steps { $t,t+1,...,t+S$ }, the LAMP autoregressively rolls out with its remeshing starting at states of $t$ for $S$ steps, and obtains the multi-step error $L^\text{evo}_S[\mathbb{I},f^\text{evo}_\theta;\hat{M}^t]$ and average state size $\mathcal{C}_S[\mathbb{I},f^\text{evo}_\theta;\hat{M}^t]$ over these $S$ steps (in Eq. (12) and (13) in main text). Then LAMP autoregressively rolls out **without** remeshing starting at state at time $t$ for $S$ steps, and obtains its multi-step error and average state size. Then the reward is computed by computing the **improvement** of LAMP, with remeshing, as shown in Eq. (11)-(13).

---

> ### Author Response · Authors · 2022-11-18
> **Official Response (4)**
>
> > Re4: Figure 4: LAMP seems to add new edges to a different regions than the ground-truth. Specifically, GT adds more edges on the cloth bending with the highest curvature, while LAMP adds the edges closer to the centre of the cloth. Can the authors provide the intuition why this might be the case and, perhaps, how to remedy this?
>
> Answer: Our RL remeshing algorithm is trained with the GNN-based learned evolution model, and the goal of remeshing is to achieve higher forward prediction accuracy (compared to the ground truth simulation results on the fine mesh) with the **current learned evolution model**. Therefore, more edges could be added to resolve the high curvature regions; more edges could also be added to some regions that may potentially lead to better forward simulation, which may not be the same as what the ground-truth adaptive remeshing does. We see that in Fig. 4 in the original submission, although LAMP’s mesh is more towards the center, it is still near the folding crease on the lower right, which LAMP finds can better improve the accuracy for the current learned evolution model. In the Fig. 8 of the updated manuscript, we also provide more example visualizations of LAMP and other baselines, showing that LAMP can learn to refine more near the folding region.

---

> ### Author Response · Authors · 2022-11-18
> **Official Response (3)**
>
> > Re3: It seems that the re-meshing process only adds 3-4 vertices on average (Table 1). I am surprised that it results in such a big change in MSE for 30 and 50 initial edges. Can the authors explain why this might be the case? Also, do you have a sense of how many edges were actually changed through re-meshing? For example, it is possible that some edges were added/collapsed, but the average number of vertices stayed the same.
>
> Answer: There are two stages of LAMP training. The first stage is pre-training the evolution model alone without remeshing, and the second stage is alternatively learning the policy with RL and finetuning the evolution model. In 1D, in the first pre-training stage, we use randomly sampled irregular meshes to train, and in the second stage, the initial mesh is a uniformly down-sampled mesh (uniformly downsampled from 100 to 50 nodes), and LAMP learns to coarsen or refine this mesh. In Table 1 of the submission, the “LAMP (no remeshing)” is using the evolution model that went through the second stage, where the evolution model may be overfit to initially uniformly downsampled mesh. In the updated manuscript, we have updated Table 1 where the “LAMP (no remeshing)” ablation is using the evolution model that has gone through the first stage of pre-training but not second stage, which is a more appropriate ablation, with a better performance. The updated Table 1 is also provided as follows:
>
> Model | Initial # vertices | Average # vertices | Error (MSE)
> :--: | :--: | :--: | :--:
> CNN | 33 | 33.0 | 2.66
> FNO | 33 | 33.0 | 2.83
> MP-PDE | 33 | 33.0 | 1.73
> LAMP (no remeshing) | 33 | 33.0 | 1.12
> LAMP (ours) | 33 | 37.6 | **1.05**
>
> Model | Initial # vertices | Average # vertices | Error (MSE)
> :--: | :--: | :--: | :--:
> CNN | 50 | 50.0 | 1.10
> FNO | 50 | 50.0 | 1.79
> MP-PDE | 50 | 50.0 | 0.98
> LAMP (no remeshing) | 50 | 50.0 | 1.08
> LAMP (ours) | 50 | 53.2 | **0.75**
>
> Model | Initial # vertices | Average # vertices | Error (MSE)
> :--: | :--: | :--: | :--:
> CNN | 100 | 100.0 | 0.81
> FNO | 100 | 100.0 | 1.39
> MP-PDE | 100 | 100.0 | 0.88
> LAMP (no remeshing) | 100 | 100.0 | **0.75**
> LAMP (ours) | 100 | 100.0 | 0.76
>
>
> (Note that in the updated Table 1, we have also added another state-of-the-art baseline, MP-PDE[1] for this 1D benchmark.)
>
> We see that the change in MSE (for the initial 33 nodes) between “LAMP (no remeshing)” and “LAMP (ours)” is smaller, with this updated ablation. Still, we see that with our full model “LAMP (ours)” outperforms “LAMP (no remeshing)”, by being able to refine and coarsen with learned adaptive remeshing.
>
> For the question of “do you have a sense of how many edges were actually changed through re-meshing”, the reviewer is correct that some edges are refined and some edges are coarsened, so that the total number of nodes may not change much. As you can see in Figure 2’s second subfigure (β=0.2), there are in total 21 edges refined and 8 edges coarsened, so the final mesh has only increased 21-8=13 edges/nodes compared to the initial mesh. Furthermore, there are 128 test trajectories, and some have net increase and some have net decrease of edges (for more example visualizations, see Figure 8 in Appendix in the updated manuscript). Therefore, on average, we don’t see a huge change in the #nodes at the end state, averaged over all test trajectories.
>
> For the scenario where the initial mesh has 100 vertices, the reason why the average #vertices for LAMP is also 100.0 is as follows. Since the ground-truth mesh has 100 vertices, we have set the constraint that LAMP cannot go beyond the spatial resolution of the ground-truth mesh. Therefore, we see that the average #vertices for our full model “LAMP (ours)” is 100.0. This shows that LAMP determines that it will remain at this mesh (since cannot refine further) and **have decided not to coarsen**, to achieve better accuracy.
>
> [1] Brandstetter, Johannes, Daniel Worrall, and Max Welling. "Message passing neural PDE solvers.", ICLR 2022

---

> ### Author Response · Authors · 2022-11-18
> **Official Response (2)**
>
> > Re2: Choice of K^max and evaluation on larger experiments. If my understanding is correct, the model performs up to K^max actions of refinement and coarsening at each step of the simulation. Intuitively, K^max should be higher on the more fine mesh, as we might need to modify more edges. How was K^max chosen?
> The current paper evaluated the simulations only up to 100 nodes, while the previous MeshGraphNet paper demonstrated the results on up to ~5000 nodes. Do the authors think that the approach can scale to this number of nodes. How do you expect the choice of K^max to change, if we increase the number of nodes to 5000? Given that the action space is N_edge^K, do the authors think that it would be feasible to train the method with such a large action space?
>
> Answer: Thanks for raising the point of scalability. Besides the 1D experiment with up to 100 nodes, in the 2D mesh-based simulation in the updated manuscript, the number of nodes is up to **235** nodes, and the average number of nodes is 123.1. Our LAMP outperforms baselines of “MeshGraphNets + GT remeshing”,  “MeshGraphNets + heuristics remeshing”, and ablation “LAMP (no remeshing)”. This shows that LAMP is able to scale to larger, more complex mesh-based simulations. Even though in appearance, the action space is on the order of $N_{\text{edge}}^{K}$, the use of Graph Neural Networks (GNNs) as policy network injects the right inductive biases to make the model complexity independent of the number of edges. More specifically, the GNN policy learns to predict the logit of actions on an edge based on the states of a local graph consisting of neighboring edges within a few hops. This policy is **shared** among all edges, independent of the total number of edges. Therefore, we expect that LAMP is able to scale to even larger mesh sizes, e.g., to 5000 nodes.
>
> The choice of $K^{\text{max}}$ is a hyperparameter which can be set by the practitioner prior to the training of the policy. We expect that for a larger initial mesh, a larger $K^{\text{max}}$ is more appropriate. We use $K^{\text{max}}$ to reduce the dimension of action space compared to the alternative design where all edges can perform their respective action (as explained in Section 3.1). We don’t see it as a major limitation. Suppose that the initial number of nodes is ~5000, and setting $K^{\text{max}}$=50 would probably suffice, since after 50 steps of rollout, LAMP can refine or coarsen up to 50*50=2500 edges, which is on the same order as 5000.

---

> ### Author Response · Authors · 2022-11-18
> **Official Response (1)**
>
> > Re1: The choice of baselines and comparison to the MeshGraphNet+remeshing. I am surprised with the choice of the baselines. The cloth mesh can be naturally represented as a graph, but the paper provides only one one graph-net baseline (LAMP with not re-meshing). Using the CNN as a baseline enforces that the initial mesh should be represented as a grid, which is quite limiting.
> Also, the paper does not provide the comparison to the previous MeshGraphNet method with re-meshing, as it is the only other method that uses re-meshing. It is not an entirely fair comparison, as MeshGraphNets use supervision on re-meshing, but it would be useful to compare which parts of the mesh are modified, and how many edges are added/collapsed between the two approaches.
>
> Answer: Thanks for raising the point of baseline comparison with MeshGraphNets + remeshing. In the updated manuscript, we have updated Table 2 by adding two more baselines. One is the baseline of “MeshGraphNets + GT remeshing”, where the “GT (ground-truth) remeshing” means that the remeshing is provided by the classical solver with Adaptive Mesh Refinement. Here we use ground-truth remeshing since it provides a lower rollout error than learned remeshing (since learned remeshing in MeshGraphNets needs to predict the sizing field for remeshing which can result in additional error), as is shown in the MeshGraphNets paper. Another baseline we have added is “MeshGraphNets + heuristics remeshing” according to reviewer tR71, where the heuristic is to refine the edge according to the local curvature. To address the point of scalability in the next question, in the updated Table 2, we now use initial states at t=10,30,50 (instead of previous t=0 only) and each rolls out for 20 steps (compared to the previous 10 steps). This has a larger initial and average mesh size, and is able to better test the scalability and longer-term prediction of our method. As another note, in the 2D mesh-based experiment, we did not use CNN as a baseline (as can be seen in the original submission), since CNN requires the mesh to be a regular grid. CNN is **only** used as a baseline in 1D with uniform grid without remeshing.
>
> Below is updated Table 2 for 2D mesh-based simulation with added baselines:
>
> Model | Initial # vertices | Average # vertices | Error (MSE)
> :--: | :--: | :--: | :--:
> MeshGraphNets + GT remeshing | 102.9 | 115.9 | 5.91e-4
> MeshGraphNets + heuristic remeshing | 102.9 | 191.9 | 6.38e-4
> LAMP (no remeshing) | 102.9 | 102.9 | 6.13e-4
> LAMP (ours) | 102.9 | 123.1 | **5.80e-4**
>
> From the updated table, we see that our LAMP outperforms both baselines and the no-remeshing ablation. Specifically, LAMP outperforms the strong baseline of “MeshGraphNets + GT remeshing”, where the remeshing is provided by the ground-truth Adaptive Mesh Refinement. This shows that LAMP can further improve upon MeshGraphNets + remeshing to learn a better remeshing policy, allowing the evolution model to evolve the system in a more faithful way. Furthermore, the “MeshGraphNets + heuristic remeshing” baseline has a larger error, showing that this intuitive baseline is suboptimal. Finally, LAMP outperforms its ablation without remeshing, showing the necessity of remeshing which can significantly reduce the prediction error.

---

### Author Response · Authors · 2022-11-18
**General Response**

We thank the reviewers for their thorough and constructive comments. We are glad that the reviewers recognize our method’s novelty (vMEa, mWzR, WPia) and significance (vMEa, mWzR, WPia, tR71). Based on the reviewers’ valuable feedback, we have performed additional experiments and updated the manuscript, which resolves the reviewers’ concerns. The major additional experiments and improvements are as follows:

1. We have added two baselines to the 2D mesh-based simulation, “MeshGraphNets + GT remeshing” (remeshing provided by solver’s Adaptive Mesh Refinement) and “MeshGraphNets + heuristics remeshing” (remeshing using heuristics based on local curvature), according to the suggestion of reviewers vMEa, mWzR, WPia, tR71. We see that our LAMP outperforms these two strong baselines, and also the ablation of no remeshing, which demonstrates the effectiveness of LAMP. For more details, see the responses to Reviewer vMEa.

2. In the updated Appendix B, we have provided a cohesive and complete discussion of the experimental setup, including the general training procedure, action space, specific treatments for 1D and 2D experiments, and details for baselines, to significantly improve the reproducibility of the work. We will also release the code upon publication of the paper, as stated in the updated Sec. 6 Reproducibility Statement.

3. We have updated Fig. 3 “error and #vertices” in which we have added markers for baseline models, showing that our LAMP significantly improves the Pareto frontier of error vs. #vertices, by achieving similar or smaller error with a much less number of vertices.
4. We have updated Table 1 for 1D nonlinear PDE benchmark. Specifically, we added another state-of-the-art model, MP-PDE, as a comparison. We see that our LAMP outperforms all baselines by a wide margin, and also outperforms the ablation of no remeshing. For more details, see the responses to Reviewer vMEa.

5. We have added Appendix C to provide more visualization of the 1D and 2D results. This gives a fuller picture of LAMP’s learned remeshing.

---

### Author Response · Authors · 2022-11-29
**Please respond to our rebuttal**

Dear reviewers,

Thanks for the constructive and thoughtful reviews. In the rebuttal, we have added two more strong baselines, provided a cohesive and complete discussion of the experimental setup, answered the reviewers' questions, and updated the manuscript. Hopefully, we have addressed all reviewers' concerns. We would appreciate it if the reviewers could read our rebuttal, update the evaluation as appropriate, and we would be happy to answer any additional questions the reviewers might have.

Thank you!

Authors

---

### Decision · Program_Chairs · 2023-01-20

**Decision:**

Accept: notable-top-25%

**Justification For Why Not Higher Score:**

N/A

**Justification For Why Not Lower Score:**

The authors have conducted additional experiments and provided ample rebuttals that changed the opinion of 3 reviewers.

**Metareview: Summary, Strengths And Weaknesses:**

This paper introduces an algorithm for simultaneous supervised learning of a dynamical model (simulation of the evolution of a physical system on a spatial mesh) and reinforcement learning of an actor-critic policy on the mesh topology (predicting which edge to refine or coarsen) w.r.t. rollouts of the dynamical model without remeshing. The dynamical model is represented as a graph neural network with latent nodes and edges and N message passings. The evolution loss includes the error due to predicting the state on a mesh that is coarsened/refined using the predicted policy, and the error due to running the dynamical/evolution model on a ground truth mesh. The policy loss is based on a reward based on the error reduction due to refinement/coarsening, and a reward based on the reduction of computation cost.
Experiments include learning the dynamics and remeshing on a 1D nonlinear PDE with forcing, and 2D mesh-based paper simulation; experiments evaluate the trade-off (coefficient beta) between remeshing cost and error reduction.

Strengths consist of the unsupervised learning of remeshing without ground-truth simulation (vMEa, mWzR), adaptively increasing the simulation accuracy by adding vertices on parts of the manifold with high curvature (vMEa, mWzR, WPia), the formulation of the problem in terms of FLOPs (mWzR, WPia), the clarity of the paper (tR71).

In the first round of reviews, weaknesses included questions about possible combinatorial explosion of the number of edge refinement/coarsening actions (vMEa), lack of comparison to MeshGraphNet with remitting (vMEA, mWzR, WPia), missing details about the experimental setup (mWzR, tR71). The authors provided very detailed rebuttals and responded by conducting more baseline experiments and explaining how the algorithm can scale to large meshes, as well as by providing lengthy clarifications and improvements to the paper. One reviewer (WPia) also noted that the method was not multi-scale as advertised, but multiresolution. Reviewer tR71 noted some missing references (corrected).

**Note From Pc:**

if the above contains the word "oral" or "spotlight" please see: "oral" presentation means -> notable-top-5% and "spotlight" means -> notable-top-25%. As stated in our emails, we are disassociating presentation type from AC recommendations

**Summary Of Ac-Reviewer Meeting:**

After the review and rebuttal process, reviewers converged on scores 5, 6, 8, 8, pushing the paper over the acceptance bar.